# Towards Realistic UAV Vision-Language Navigation: Platform, Benchmark, and Methodology

**Xiangyu Wang**[1*]**, Donglin Yang**[1*]**, Ziqin Wang**[1*]**, Hohin Kwan**[1]**, Jinyu Chen**[1]**,
Wenjun Wu**[2]**, Hongsheng Li**[3,4]**, Yue Liao**[3,4†]**, Si Liu**[1†]
[1] Institute of Artificial Intelligence, Beihang University,
[2] Hangzhou International Innovation Institute of Beihang University,
[3] MMLab, CUHK, [4] Centre for Perceptual and Interactive Intelligence
{wangxiangyu0814,yangdonglin,liusi}@buaa.edu.cn, liaoyue.ai@gmail.com

## Abstract

Developing agents capable of navigating to a target location based on language instructions and visual information, known as vision-language navigation (VLN), has attracted widespread interest. Most research has focused on ground-based agents, while Unmanned Aerial Vehicle (UAV)-based VLN remains relatively underexplored. Recent efforts in UAV VLN predominantly adopt ground-based VLN settings, relying on predefined discrete action spaces and neglecting the inherent disparities in agent movement dynamics and the complexity of navigation tasks between ground and aerial environments. To address these disparities and challenges, we propose solutions from three perspectives: platform, benchmark, and methodology. To enable realistic UAV trajectory simulation in VLN tasks, we propose the TRAVEL platform, which features diverse environments, realistic flight control, and extensive algorithmic support. We further construct a target-oriented VLN dataset consisting of approximately 12k trajectories on this platform, serving as the first dataset specifically designed for realistic UAV VLN tasks. To tackle the challenges posed by complex aerial environments, we propose an assistant-guided UAV object search benchmark called UAV-Need-Help, which provides varying levels of guidance information to help UAVs better accomplish realistic VLN tasks. We also propose a UAV navigation LLM that, given multi-view images, task descriptions, and assistant instructions, leverages the multimodal understanding capabilities of the MLLM to jointly process visual and textual information, and performs hierarchical trajectory generation. The evaluation results of our method significantly outperform the baseline models, while there remains a considerable gap between our results and those achieved by human operators, underscoring the challenge presented by the UAV-Need-Help task. The project homepage can be accessed at https://prince687028.github.io/Travel.

## 1 Introduction

Constructing embodied agents capable of understanding human commands remains a long-term objective in the field of artificial intelligence. Among these (Qi et al., 2020; Ku et al., 2020; Shridhar et al., 2020; Shen et al., 2021), visual-language navigation (VLN)—navigating to a target location based on language instructions and visual information—has garnered significant research interest. Current research in VLN focuses primarily on ground-based agents (Krantz et al., 2020; Blukis et al., 2018), while UAV-based VLN has received comparatively less attention. This area presents a wealth of application scenarios, and due to the notable differences in action space and observations between UAVs and ground-based agents, it represents a valuable area for research.

---

[*]Equal contribution.
[†]Corresponding author.

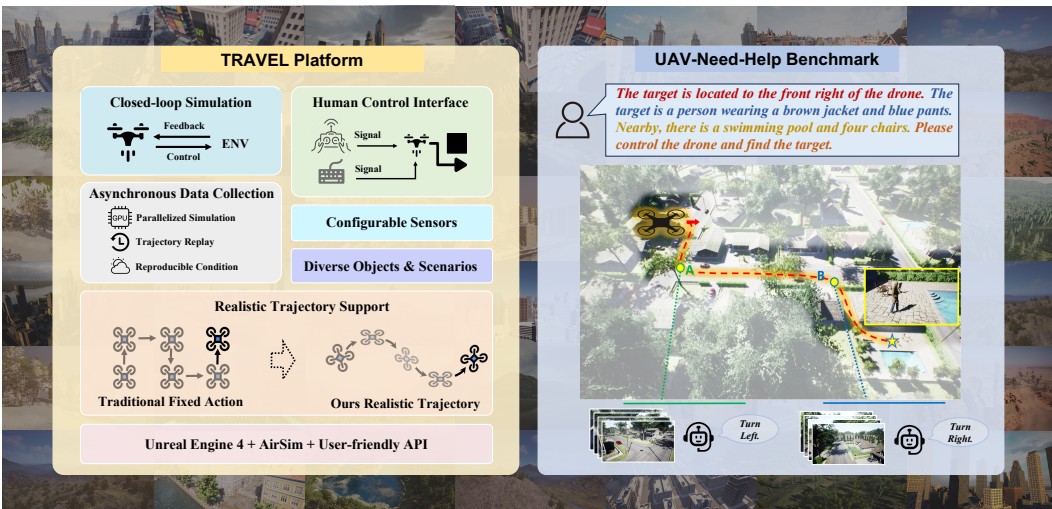

Figure 1: We propose a realistic UAV simulation platform and a novel UAV-Need-Help benchmark. The TRAVEL platform focuses on realistic UAV VLN tasks, integrating diverse environmental components, realistic flight simulations, and algorithmic support. The UAV-Need-Help benchmark introduces an assistant-guided UAV object search task, where the UAV navigates to a target object using object descriptions, environmental information, and guidance from assistants.

Recent UAV VLN benchmarks (Liu et al., 2023b; Fan et al., 2022; Lee et al., 2024) typically adopt ground-based VLN settings, relying on a fixed set of discrete actions. However, we argue that a significant gap exists between ground-based movement and UAV flight characteristics, along with the distinct operational environments, making a direct transfer of ground-based approaches to UAVs insufficient for precise aerial navigation. The primary differences and challenges are manifested in two aspects: 1) *Mismatch in agent movement dynamics.* Ground-based agents (Anderson et al., 2018; Yan et al., 2019; Shah & Levine, 2022) typically move along a horizontal plane, which makes it straightforward to plan navigation using discrete actions such as horizontal movement and rotation. In contrast, UAVs can operate freely in three-dimensional airspace. As shown in Fig. 1, traditional methods attempt to define UAV movement using fixed action sets, which incorporate basic directions such as up, down, and horizontal movements. These approaches oversimplify UAV control and fail to capture realistic flight dynamics, as UAVs often perform maneuvers like pitching up, diving, and rolling simultaneously to achieve spatial movement. UAV trajectories are inherently continuous and difficult to decompose into discrete actions, leading to unrealistic navigation when such simplifications are applied. 2) *Disparity in navigation task complexity*. UAVs frequently operate in diverse outdoor, open environments, where navigation paths are often long and complex. Unlike ground-based agents (Anderson et al., 2018) that navigate using well-defined descriptions of target objects, UAVs must deal with obstructed views and shifting perspectives due to their high maneuverability. Therefore, relying solely on target descriptions is inadequate for precise localization and navigation in such complex, dynamic scenarios. To address these disparities and associated challenges, we propose a UAV simulation platform **TRAVEL**, a target-oriented VLN dataset, a novel benchmark **UAV-Need-Help**, along with a UAV navigation LLM to create a more realistic framework for UAV vision-language navigation which supports continuous trajectories and complex scenarios.

To narrow the gap to realistic UAV VLN tasks, a crucial advancement lies in replacing fixed action sequences with continuous flight trajectories. To facilitate this, we introduce **TRAVEL**, a simulation platform **T**owards **R**ealistic **A**erial **V**ision and **E**xploratory **L**anguage navigation, offering realistic environments, authentic flight simulation, and extensive algorithmic support. As shown in Fig. 1, we utilize UE4's realistic rendering capabilities, integrate 22 scenarios and 89 objects, and provide APIs for object placement and scenario configuration. We integrate AirSim plugin to translate trajectory sequences into continuous paths with realistic flight dynamics and support continuous flight control using real flight signals via remote controllers or designed APIs. Building on the features above, we develop a parallel realistic trajectory collection framework and a closed-loop simulation framework, providing comprehensive algorithmic support for UAV tasks.

Table 1: Comparison of existing VLN datasets based on the DoF of navigation action space and dataset scale. Our UAV dataset incorporates 6 DoF trajectories for realistic VLN tasks. $N_{traj}$ and $N_{vocab}$ represent the number of collected trajectories and vocabulary size, respectively.

| Dataset | DoF | Kinematics | $N_{traj}$ | $N_{vocab}$ | Traj Len. | Intr Len. | Actions | Environment |
|---------|-----|-----------|-----------|-------------|-----------|-----------|---------|-------------|
| R2R | - | ✗ | 7189 | 3.1K | 10 | 29 | 5 | Matterport3D |
| TouchDown | 2 | ✗ | 9326 | 5K | 310 | 90 | 35 | Google Street View |
| LANI | 2 | ✗ | 6000 | 2.3K | 16 | 57 | 116 | CHALET |
| AVDN | 3 | ✗ | 6269 | 3.3K | 144 | 89 | 7 | xView |
| AerialVLN | 4 | ✗ | 8446 | 4.5K | 661 | 83 | 204 | AirSim |
| CityNav | 4 | ✗ | 32637 | 6.6K | 545 | 26 | - | SensatUrban |
| Ours | 6 | ✓ | 12149 | 10.8K | 255 | 104 | 264 | TRAVEL |

To mitigate the scarcity of realistic UAV trajectory data for VLN tasks, We leverage the unique features of the TRAVEL platform to construct a target-oriented VLN dataset, which is the first realistic UAV VLN dataset to incorporate 6 degrees of freedom (DoF) motion, accurately capturing the complex flight dynamics of UAVs. Human annotators conducted continuous flights on the TRAVEL platform, regularly recording UAV states during annotation and asynchronously collecting additional sensor data to obtain navigation trajectories. We utilize GPT-4 to generate target descriptions and follow up with manual quality checks to generate high-quality navigation instructions, which result in a total of approximately 12k trajectory-instruction pairs.

The high maneuverability of UAVs and the complex aerial environments pose significant challenges for UAV object search task relying solely on descriptive instructions. To support UAV to accomplish such tasks, we enhance UAV navigation capabilities by providing additional information and accordingly propose the **UAV-Need-Help** benchmark. As shown in Fig. 1, we establish an assistant that provides action guidance to UAVs in specific scenarios and categorize the assistant into three types based on the varying levels of guidance. The UAV performs the task based on the initial target description, environmental information, and instructions from the assistant.

To tackle the challenges presented by the UAV-Need-Help benchmark and enhance the efficiency of UAVs in object search tasks, we propose a UAV navigation LLM. We leverage MLLM's understanding (Li et al., 2024; Han et al., 2024) and decision-making Zhang et al. (2024) capabilities to produce hierarchical outputs for long-distance and fine-grained trajectories. Additionally, we propose a backtracking sampling-based data aggregation strategy to enhance environmental understanding and obstacle avoidance capabilities. Comprehensive closed-loop evaluations across various settings validate the effectiveness of our approach.

We hope that the proposed platform, benchmark, and methodology can promote research and development in VLN tasks based on continuous and realistic UAV trajectories, thereby facilitating the transfer of UAV VLN systems to real-world applications.

## 2 RELATED WORK

**Embodied Simulator.** Simulation platforms are crucial for intelligent systems research. For ground-based simulations, platforms like Habitat (Puig et al., 2024), Matterport3D (Chang et al., 2017) and Google Street View (Anguelov et al., 2010) have been pivotal simulated realistic indoor and outdoor scenes, significantly propelled research by improving data collection and algorithm evaluation in complex scenarios. However, UAV simulation platforms are still in their early stages of development. Some existing platforms, such as Gazebo (Koenig & Howard, 2004), provide a foundation for UAV control and navigation, but they still exhibit limitations in terms of realism, scalability, and adaptability to various scenarios. The xView platform (Lam et al., 2018) only offers remote sensing satellite images which are unsuitable for low-altitude navigation tasks. Recently, some studies (Liu et al., 2023b; Ma et al., 2020; Haley et al., 2023; Gill et al., 2021; Madaan et al., 2020; Alvey et al., 2021) have combined UE with AirSim to develop high-fidelity imaging platforms. However, these platforms typically lack algorithmic support for continuous trajectory VLN tasks. In contrast, our platform offers realistic environments, authentic flight simulation, and extensive algorithmic support, providing a foundation for realistic UAV VLN research.

**Vision-Language Navigation Datasets.** UAV VLN is an emerging task recently formed from the expansion of outdoor VLN tasks. Table 1 presents a comparison between representative VLN datasets. Beginning with indoor navigation, R2R dataset (Anderson et al., 2018) establishes a foundation for grounded VLN. Inspiring works (Vasudevan et al., 2020; Zhu et al., 2021; Chen et al., 2019; Misra et al., 2018) such as TOUCHDOWN, LANI have employed verbal navigation instructions to address the challenges of outdoor long-range navigation. Moving beyond ground-level navigation, AVDN (Fan et al., 2022) collects aerial navigation trajectories with human dialogues. AerialVLN (Liu et al., 2023b) and CityNav (Lee et al., 2024) propose VLN for UAVs and collect aerial trajectory datasets of discrete 4-DOF UAV action. In previous datasets, UAV trajectories were collected by simply modifying the UAV's position, which significantly deviates from real UAV flights. Based on our platform, we collect a target-oriented realistic UAV VLN dataset with continuous 6-DOF trajectories to enhance the realism of UAV navigation tasks.

**LLM-based Navigation Agent.** The application of LLM-based methodologies has led to significant advancements in developing navigation agents. LM-Nav (Shah et al., 2022) propose a goal-conditioned robotic navigation method that enables robust generalization to real-world outdoor environments without language-annotated data. LMDrive (Shao et al., 2024) further progressed the field by introducing a language-guided, end-to-end autonomous driving framework that integrates multi-modal sensor data, facilitating effective human-robot interaction in challenging urban scenarios. Some studies (Vemprala et al., 2023; Zhong et al., 2024) have utilized the zero-shot capabilities of LLMs to generate UAV navigation code, while (Lee et al., 2024) combined LLMs for coordinate optimization of drone target points, validating the basic capabilities of LLMs in drone navigation tasks. To further expand the application scenarios for drones, we proposed a UAV navigation LLM composed of hierarchical trajectory generation models in solving the object search task.

## 3 TRAVEL SIMULATION PLATFORM

As illustrated in Fig. 1, the TRAVEL simulation platform is a fully open-source platform devoted to realistic UAV VLN tasks, integrating three modules: environment composition, flight simulation, and algorithmic support to achieve comprehensive functionality.

### 3.1 ENVIRONMENT COMPOSITION

**Diverse Scenario Resources.** TRAVEL offers a wide range of scenarios and achieves high-fidelity visual effects through the advanced graphical rendering capabilities of UE4. We integrate high-quality scenarios from online repositories and urban scenarios from the CARLA simulator (Dosovitskiy et al., 2017), resulting in the TRAVEL platform featuring 22 distinct scenarios, including urban, rural, and natural landscapes, as detailed in Appendix A. The platform can also simulate dynamic environments, such as vegetation swaying and variations in lighting. This further enhances the platform's realism, reducing the gap when transitioning to real-world environments.

**Tailored Object Assets.** TRAVEL platform features a wide variety of standalone object assets, including humans, vehicles, animals, road signs, tables and other items suited for urban and natural environments. Users can place objects in scenarios using different methods based on task requirements. In addition to UE4's built-in scenario editor, we support the automatic placement method at runtime. The method is implemented through the TRAVEL API, where we have pre-defined feasible regions within each scenario and classified various object categories. Annotators can utilize the object placement API to select the object type and place it in appropriate and suitable areas.

### 3.2 FLIGHT SIMULATION

**Realistic UAV Flight Control.** TRAVEL platform integrates the AirSim plugin to achieve realistic UAV flight simulation, enabling more precise flight control. Our platform can utilize the flight control API to achieve physics-based UAV maneuvers with a 6 DoF trajectory representation. The pose of the UAV at each time is represented as $P = \{x, y, z, \theta, \phi, \psi\}$, where $(x, y, z)$ represent the position coordinates, and $(\theta, \phi, \psi)$ denote the pitch, roll, and yaw angles respectively. The pose can be obtained at any time, allowing highly accurate simulation of UAV movement.

**Configurable UAV Sensors.** TRAVEL platform supports the simulation of various sensor payloads during UAV flight, including IMU, RGB and depth cameras, LiDARs and GPS. The platform initially configures the UAV with RGB and depth cameras covering front, rear, left, right, and downward views, along with a LiDAR sensor set up with a 360-degree horizontal field of view. Users can add necessary payloads and adjust detailed sensor configurations based on specific requirements, such as modifying image resolution to optimize response time.

**Human Control Interface.** We integrate AirSim's hardware-in-the-loop simulation capability and use PX4 to support remote controller operation. However, the control link in AirSim can be relatively complex for general users. Therefore, we develop simplified APIs that allows users to simulate remote controller input via the keyboard, supporting functions such as data collection, and enabling operation in both manual mode and position mode.

### 3.3 Algorithmic Support

**Data Collection Tools.** TRAVEL platform has contained a data collection framework to tackle the challenge of limited UAV training data. Collecting data from multiple sensors requires a certain amount of time, which can affect the continuity of annotation. Therefore, we have implemented an asynchronous collection method that initially gathers UAV attitude information at equal time intervals, followed by the data collection of sensors in the background.

**Closed-loop Simulation.** The platform provides an extended interface for the UAV navigation model, allowing for flexible integration of model outputs to control drone flight and real-time feedback of environmental information to the model. Moreover, we have implemented a dataset aggregation (DAgger) method to enhance model training and proposed a backtracking policy. When the UAV encounters a collision, it is reverted to its previous position and uses teacher action to perform the next step, allowing it to recover from the collision and produce longer navigation trajectories.

**Parallelization.** TRAVEL platform employs an environment parallelization strategy, allowing multiple simulated environments to run concurrently to enhance the efficiency of data collection and closed-loop evaluation. With 8 NVIDIA A100 GPUs, the simulation of a single UAV achieves a performance boost of 16 times, reaching frame rates of between 160 and 1600 fps, depend on the amount of data captured by UAV payloads.

## 4 Target-Oriented Realistic UAV VLN Dataset

We construct a target-oriented realistic UAV VLN dataset using the proposed TRAVEL platform, which is the first dataset accurately capturing complex flight dynamics for UAV VLN tasks. As shown in Fig. 2 (a), we introduce this dataset in two aspects: data collection and data analysis.

### 4.1 Data Collection

**Description Collection.** In the UAV object search task, the target descriptions consist of three key components: 1) Target direction, indicating the relative position between the target and UAV's initial pose; 2) Object description, detailing the visual features of the target; and 3) Environmental information, describing the surrounding spatial context. During the description collection process, objects can be placed in the feasible regions of the scene using the TRAVEL object placement API. Object descriptions and environment information are then obtained by positioning a camera above the object to capture images from five different views (front, back, left, right, and down), followed by generating textual descriptions using GPT-4 (Achiam et al., 2023), with the prompts detailed in Appendix B. Human experts review and refine the generated content to maintain data quality, removing any inaccuracies or hallucinations.

**Asynchronous Trajectory Collection.** Human common sense in target searching provides valuable guidance for UAVs to learn effective search strategies. During the trajectory collection process, human experts utilize the human control interface of the TRAVEL platform to manually control the UAV according to the given instructions to search for the target. To minimize operational discontinuity caused by sensor data storage latency, only UAV states are recorded at fixed time intervals during the flight. After completing the trajectory collection, sensor data is acquired through the parallel data collection framework of TRAVEL. The trajectory recording terminates when the UAV

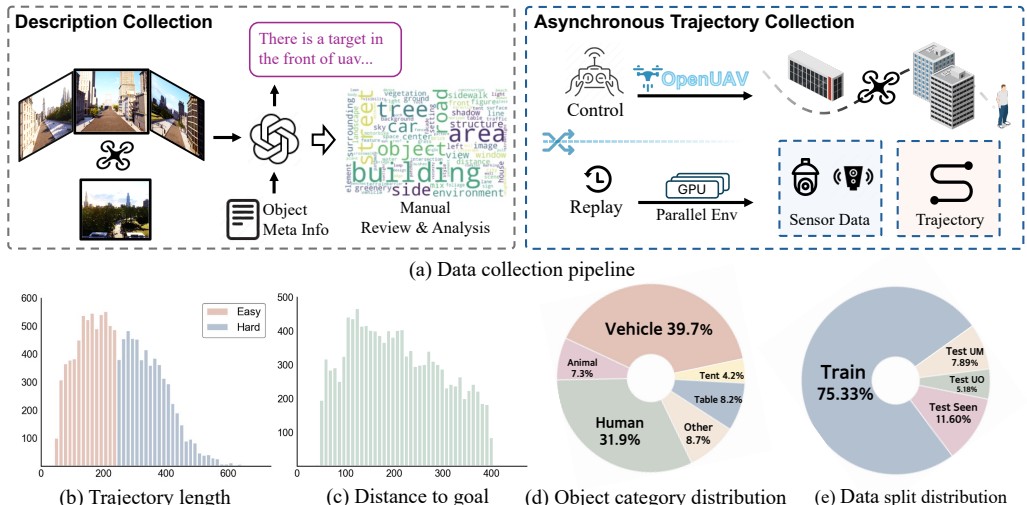

Figure 2: Overview of our dataset construction and statistical analysis. (a) Data collection pipeline for generating high-quality target descriptions and realistic UAV trajectories. (b) - (e) Statistical analysis of the dataset, covering trajectory lengths, task distances, object categories, and dataset splits. In (e), UM and UO represent Unseen Map and Unseen Object, respectively.

approaches within 5 meters of the target, and any trajectory is discarded if a collision is detected. Ultimately, a total of 12,149 valid trajectories are obtained.

## 4.2 DATA ANALYSIS

**Trajectory Analysis.** As illustrated in Fig. 2 (b), the UAV-Need-Help dataset contains a total of 12,149 trajectories, where trajectories shorter than 250 meters are classified as *easy*, and those exceeding 250 meters are classified as *hard*. The diversity in trajectory lengths ensures the challenge and complexity of the tasks. Fig. 2 (c) illustrates that the distance to the targets ranges from 50 meters to 400 meters, representing the spatial scale of the environment.

**Description Analysis.** The most frequently occurring descriptions are illustrated in Fig. 2 (a), including building, tree, and car. These descriptions provide contextual information for UAVs, enhancing their capability to estimate object locations through visual cues and thereby accurately find the target. Fig. 2 (d) shows the target object set consists of 89 distinct categories for search tasks, encompassing vehicles, humans, animals and other objects.

**Dataset Split.** As shown in Fig. 2 (e), to comprehensively evaluate the model's performance on both seen and unseen environments, and to analyze its ability to generalize to new maps and objects, we divide the dataset into four subsets: *Train*, *Test Seen*, *Test Unseen Map*, and *Test Unseen Object*. Each test subset is further divided into *easy* and *hard* categories, consistent with the criteria mentioned above. Specifically, the data distribution for each subset is as follows:

- *Train* - 9152 trajectories with 76 objects across 20 scenes as the training set.
- *Test Seen* - 1410 trajectories generated using objects and scenes seen in the training set.
- *Test Unseen Map* - 958 trajectories with 2 scenes unseen in the training set.
- *Test Unseen Object* - 629 trajectories with 13 objects unseen in the training set.

## 5 UAV-NEED-HELP BENCHMARK

### 5.1 TASK FORMULATION

We propose an assistant-guided UAV object search task named UAV-Need-Help, where the UAV navigates to the target object following the target description, environment information and guidance from the assistant. Fig. 1 provides an illustration of the UAV-Need-Help task.

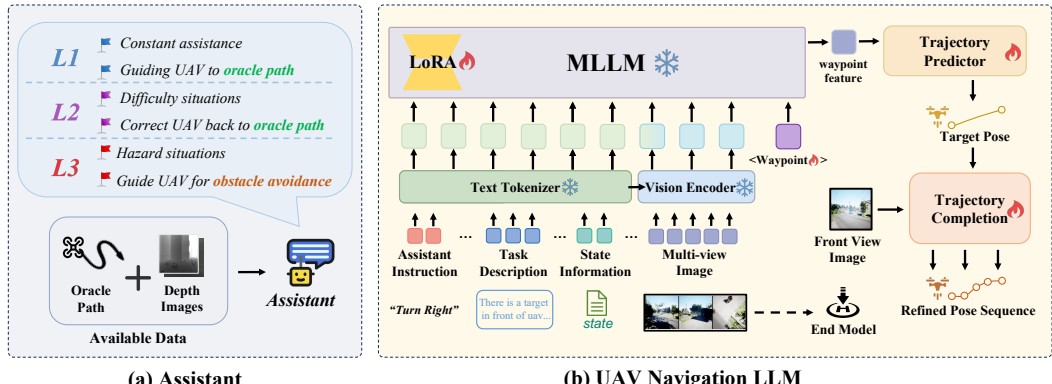

Figure 3: Overview of the assistant mechanism and UAV Navigation LLM framework. (a) Three different assistant settings for providing varying levels of guidance. (b) UAV Navigation LLM framework: The instructions, multi-view images, and a learnable query are encoded into the MLLM, where the query extracts features to predict a long-distance target pose. This pose is then refined using front-view inputs by a trajectory completion model to generate fine-grained trajectories.

Formally, in each episode, the UAV starts at an initial position with posture $P_0$ and receives a target description $I$ that specifies the target direction, object features, and its surrounding environment. At each time step, the UAV obtains its state $S$ (position, posture, velocity), along with RGB images $R$ and depth images $D$ from five perspectives: front, left, right, rear, and below. An assistant monitors its status and provides additional instructions $I'$ to suggest flight strategies when needed. In the closed-loop simulation, the UAV navigation model predicts a 6 DoF trajectory sequence based on $\{I, S, R, D\}$. Using the TRAVEL platform's flight API, the UAV navigates to each predicted position with a 6 DoF attitude $\{x', y', z', \theta', \phi', \psi'\}$ while adhering to its flight dynamics and updating its observations. The task is successful if the UAV lands within a 20-meter radius of the target.

## 5.2 ASSISTANT MECHANISM

As previously mentioned, the inherent complexity and dynamic nature of aerial environments, with obstructed views and shifting perspectives, make basic target descriptions insufficient for UAV object searching. Thus, we introduce assistants to aid UAVs in this task, defining three distinct tiers based on the level of guidance provided, as shown in Fig. 3 (a).

- **L1** assistant provides high-frequency guidance closely aligned with the ground truth (GT) trajectory. It continuously calculates the trajectory point in the oracle path that is closest to the UAV's current position, then identifies the appropriate actions based on the UAV's orientation and the direction toward the nearest GT point. These actions may include cruising, turning, or landing. This setting ensures that the UAV stays on the correct path at all times.
- **L2** assistant intervenes when the UAV encounters difficulties, providing low-frequency corrections to steer it back toward the ground truth trajectory. It activates only when a collision risk is identified through depth maps or when the UAV deviates too far from the GT path, offering corrective actions to guide the UAV back to the desired trajectory.
- **L3** assistant only provides obstacle avoidance assistance when the UAV is in a hazardous scenario. It determines the distance between the UAV's current position and obstacles using depth maps, and when proximity to an obstacle is detected, it issues avoidance commands to prevent collisions.

## 6 UAV NAVIGATION LLM

UAV navigation LLM is a multimodal LLM capable of handling various input types, including images and texts. We first tokenize the multi-model input, where vision tokens are aligned with the language space. These tokens are then concatenated and fed into the LLM, with Vicuna-7B serving as the base model. Utilizing the features obtained by LLM, a hierarchical trajectory decoder generates both the UAV's next target pose $P_{target}$ and refined pose sequence $\{P_{traj}^0, P_{traj}^1 ... P_{traj}^N\}$,

where $N$ represents the length of this trajectory. The overview architecture is shown in Fig. 3 (b). Additionally, we have designed a backtracking sampling-based data aggregation to expand the dataset, thereby enhancing models' obstacle avoidance capabilities in complex scenarios.

## 6.1 Hierarchical Trajectory Generation

**Multimodal Tokenization.** Given the task description $I_{task}$, assistant instructions $I_{instr}$ and the UAV state information $I_{state}$, we tokenize them using a pre-trained language tokenizer (Li et al., 2024) to obtain corresponding tokens $T_{instr}$. For the multi-view images, we utilize a EVA-CLIP (Sun et al., 2023) and a Q-former structure to extract the visual features. Each image is converted into a set of tokens, consisting of 1 context token that captures global features and 16 content tokens that represent local details through grid pooling. Finally, we concatenate the three types of tokens: $T_{img}$ for image tokens, $T_{task}$ for task description tokens, and $T_{instr}$ for instruction tokens, to construct the multi-modal input token sequence, denoted as $T_{input} = < T_{img}, T_{task}, T_{instr} >$. The details of the multimodal input can be referenced in Appendix D.

**Hierarchical Trajectory Decoder.** The decoder is the core component that translates the input into the UAV's next step target pose. To enable complex trajectory planning within dynamic environments, we employ a hierarchical structure with two levels: a high-level MLLM-based trajectory decoder and a fine-grained path decoder.

The MLLM-based trajectory decoder utilizes a special learnable trajectory token $T_{wp}$ as input to the LLM, which helps to extract trajectory-specific features $Q_{wp}$. These features are fed into a multilayer perceptron (MLP) to decode the target pose $P_{target}$, allowing the model to plan UAV trajectories based on environmental data and detailed language instructions.

The fine-grained path decoder generates trajectory details to improve navigation efficiency. It takes visual tokens $T_{img}^f$ encoded from the front-view image using pre-trained ViT and combines $T_{img}^f$ with processed poses derived from the MLLM-based decoder. The concatenated features are passed through an MLP to produce fine-grained trajectories $P_{traj}$. The overall process is defined as follows:

$$P_{traj} = \text{MLP}\left(T_{img}^f, f_{target}\left(\text{MLLM}(< T_{input}, Q_{wp} >)\right)\right),$$

where $f_{target}$ represents the alignment process of target poses from the MLLM output with the dimensions of visual tokens. To further improve UAV landing accuracy near the target, a Grounding DINO model (Liu et al., 2023a) is utilized for detection. Once the target object is identified in multi-view images, the navigator initiates the landing process.

## 6.2 Backtracking Sampling-based Data Aggregation

We implement a DAgger (Ross et al., 2011) module for continuous and realistic UAV trajectories. During data collection, the module samples from both predicted trajectories and reference paths provided by the teacher model. The teacher model selects the closest ground truth position to the UAV's current location and computes the next trajectory, which guides the UAV in VLN tasks. Considering that collisions often occur when simulating UAV flight, it becomes challenging to collect complete trajectories or those containing sufficient obstacle avoidance knowledge. Thus, we propose a backtracking sampling mechanism, where, if the UAV takes an action based on the model's output that results in a collision, it is reverted to its state from two frames earlier, restoring its pose, velocity, and other attributes. It will then follow the trajectory given by the teacher model, helping the UAV to avoid collisions and stay on track. In this way, we can obtain more successful obstacle-avoidance trajectories, thereby enhancing the navigation capability of the model.

## 7 Experiments

### 7.1 Experimental Setup

**Evaluation Metrics.** We adapt evaluation metrics commonly used in VLN (Anderson et al., 2018; Krantz et al., 2020), including success rate (SR), oracle success rate (OSR), success weighted by path length (SPL), and navigation error (NE). SR measures the percentage of tasks where the UAV

Table 2: Results on Test Seen Set across different assistant levels. DA refers to a model trained using backtracking sampling-based data aggregation.

| Method | Assistant | Full | | | | Easy | | | | Hard | | | |
|---|---|---|---|---|---|---|---|---|---|---|---|---|---|
| | | NE↓ | SR↑ | OSR↑ | SPL↑ | NE↓ | SR↑ | OSR↑ | SPL↑ | NE↓ | SR↑ | OSR↑ | SPL↑ |
| Random | L1 | 222.20 | 0.14 | 0.21 | 0.07 | 142.07 | 0.26 | 0.39 | 0.13 | 320.12 | 0.00 | 0.00 | 0.00 |
| Fixed | L1 | 188.61 | 2.27 | 8.16 | 1.40 | 121.36 | 3.48 | 11.48 | 2.14 | 270.69 | 0.79 | 4.09 | 0.49 |
| CMA | L3 | 140.93 | 4.89 | 11.56 | 4.41 | 83.58 | 7.35 | 17.81 | 6.53 | 210.91 | 1.89 | 3.94 | 1.83 |
| CMA | L2 | 141.55 | 7.02 | 15.39 | 6.54 | 87.77 | 9.55 | 19.87 | 8.74 | 207.18 | 3.94 | 9.92 | 3.94 |
| CMA | L1 | 135.73 | 8.37 | 18.72 | 7.90 | 84.89 | 11.48 | 24.52 | 10.68 | 197.77 | 4.57 | 11.65 | 4.51 |
| Ours | L3 | 146.32 | 6.31 | 15.39 | 5.10 | 93.15 | 9.55 | 21.94 | 7.32 | 215.85 | 2.36 | 7.40 | 2.17 |
| Ours | L2 | 120.57 | 12.98 | 37.38 | 11.30 | 76.89 | 17.55 | 43.48 | 15.01 | 186.22 | 7.40 | 29.92 | 6.76 |
| Ours | L1 | 106.28 | 16.10 | 44.26 | 14.30 | 68.78 | 18.84 | 47.61 | 16.39 | 152.04 | 12.76 | 40.16 | 11.76 |
| Ours-DA | L1 | **98.66** | **17.45** | **48.87** | **15.76** | **66.40** | **20.26** | **51.23** | **18.10** | **138.04** | **14.02** | **45.98** | **12.90** |
| Human | L1 | 14.15 | 94.51 | 94.51 | 77.84 | 11.68 | 95.44 | 95.44 | 76.19 | 17.16 | 93.37 | 93.37 | 79.85 |

successfully reaches the target. OSR measures whether the UAV reaches any location along the optimal trajectory, even if it doesn't exactly reach the final destination. SPL evaluates both the success rate and the efficiency of the path taken, rewarding shorter, more optimal paths. NE calculates the average distance between the UAV's final position and the target.

**Comparison Baselines.** We compare the following baselines against our method on the UAV-Need-Help task. 1) Random: The UAV randomly selects trajectory poses without any structured planning or guidance. This method is employed to illustrate the extent of the solution space. 2) Fixed Action: The UAV maps the assistant's instructions into predefined fixed actions. For example, *cruise* means moving forward by 5 meters, while *turn left* results in a 30-degree turn followed by a 5-meter forward movement. 3) Cross-Modal Attention (CMA): The CMA model (Anderson et al., 2018) is commonly used in grounded VLN tasks and employs a bi-directional LSTM to simultaneously process image inputs and instruction comprehension. To adapt to our task setting, we modify its recurrent predictor to output a set of trajectories instead of traditional navigation actions.

## 7.2 QUANTITATIVE RESULT

**Comparison with Baselines.** Table 2 shows that our method outperforms across all metrics at different difficulty levels on the test seen set. At various assistant levels, the SR metric improves by an average of 5% over the CMA model. This demonstrates that our MLLM-based hierarchical trajectory generation approach enhances scene understanding and produces more accurate and adaptable pose sequences, ultimately improving decision-making and overall performance.

Table 3: Generalization capabilities across different test sets with L1 assistant, where UO and UM represent the Test Unseen Object Set and the Test Unseen Map Set, respectively.

| Method | Test Set | Full | | | | Easy | | | | Hard | | | |
|---|---|---|---|---|---|---|---|---|---|---|---|---|---|
| | | NE↓ | SR↑ | OSR↑ | SPL↑ | NE↓ | SR↑ | OSR↑ | SPL↑ | NE↓ | SR↑ | OSR↑ | SPL↑ |
| Random | UO | 260.14 | 0.16 | 0.16 | 0.16 | 174.10 | 0.48 | 0.48 | 0.48 | 302.96 | 0.00 | 0.00 | 0.00 |
| Fixed | UO | 212.84 | 3.66 | 9.54 | 2.16 | 151.66 | 6.70 | 13.88 | 3.72 | 243.29 | 2.14 | 7.38 | 1.38 |
| CMA | UO | 155.79 | 9.06 | 16.06 | 8.68 | 102.92 | 14.83 | 22.49 | 13.90 | 182.09 | 6.19 | 12.86 | 6.08 |
| Ours | UO | **118.11** | **22.42** | **46.90** | **20.51** | **86.12** | **24.40** | **49.28** | **22.03** | **134.03** | **21.43** | **45.71** | **19.75** |
| Random | UM | 202.98 | 0.00 | 0.00 | 0.00 | 158.46 | 0.00 | 0.00 | 0.00 | 265.88 | 0.00 | 0.00 | 0.00 |
| Fixed | UM | 180.47 | 0.52 | 2.61 | 0.39 | 132.89 | 0.89 | 4.28 | 0.67 | 247.72 | 0.00 | 0.25 | 0.00 |
| CMA | UM | 141.68 | 2.30 | 10.02 | 2.16 | 102.29 | 3.57 | 14.26 | 3.33 | 197.35 | 0.50 | 4.03 | 0.50 |
| Ours | UM | **138.80** | **4.18** | **20.77** | **3.84** | **102.94** | **4.63** | **22.82** | **4.24** | **189.46** | **3.53** | **17.88** | **3.28** |

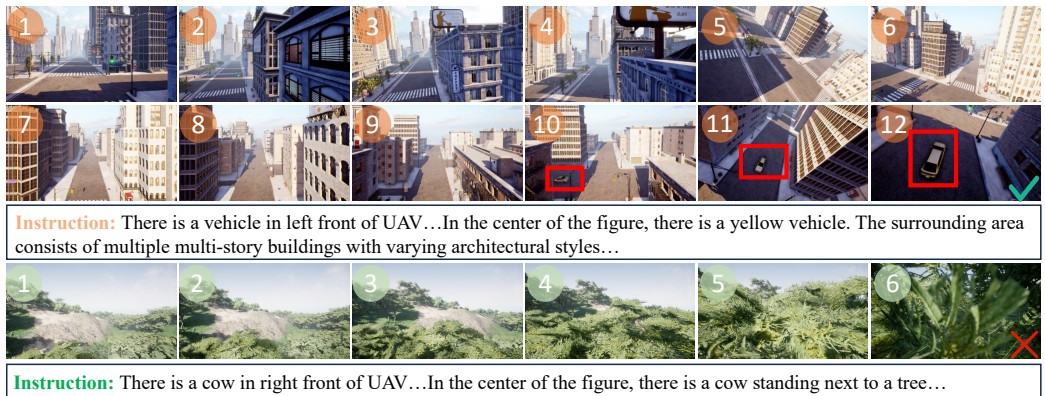

Figure 4: Visualization of object search results of our method. First two rows demonstrate our UAV successfully follows the instruction. Notably, the third to fifth images depict the drone executing a turning maneuver, resulting in a change in the drone's perspective. The third row illustrates a failed example, depicting a collision with trees in a forest scenario.

The classical baseline model CMA faces several challenges due to limitations in model size and task complexity and performs poorly on multiple metrics. The Random and Fixed methods struggle to complete the tasks, demonstrating that achieving the target search tasks without understanding instructions and visual information is extremely difficult. We further evaluate our model trained with backtracking sampling-based data aggregation, which shows an improvement in SR compared to the original model, indicating this method has enhanced the UAV's navigation capability.

Additionally, we evaluate the performance of humans operating UAVs with guidance from L1 assistant. As shown in Table 2, humans achieve a high success rate but sometimes take longer paths. This suggests that while humans make correct decisions, they may choose more cautious or exploratory routes, leading to slightly less efficient path planning.

**Comparison under Different Assistant Level.** Under the continuous guidance of the L1 assistant, our method achieves the highest success rate. As the assistant level increases, the UAV agent needs to rely more on its navigation planning capabilities to complete tasks. The results show that both our method and the CMA method experience a decline in success rate and other metrics at higher levels of assistance, demonstrating that long-term VLN for UAVs is extremely challenging.

**Generalization on Unseen Cases.** Table 3 shows the performance on unseen datasets, highlighting our method's superior zero-shot capability and its adaptability to new scenarios. Our method demonstrates a slightly higher success rate in the unseen object test set than in the seen set. This phenomenon can be attributed to the inherent generalization capabilities of the Grounding DINO module (Liu et al., 2023a) used in our approach and the relatively simpler nature of the data. We have still partitioned this dataset for future research aimed at models that do not require external detectors. In the unseen scene test set, results of the fixed action method reflect the complexity of the environment through its low success rate. Our method shows a noticeable decline in the unseen scene test set, yet still outperforms other methods.

**Performance Scalability.** Table 4 reports the model's performance across various quantities of training data with L1 assistant. The results show that as the data volume increases, the model performs better, indicating that a larger and more diverse dataset could enhance the model's understanding and decision-making capabilities.

Table 4: Performance scalability with varying amounts of training data.

| Data Amount | NE↓ | SR↑ | OSR↑ | SPL↑ |
|---|---|---|---|---|
| 10% | 126.50 | 10.78 | 31.21 | 9.68 |
| 50% | 122.63 | 13.19 | 37.16 | 11.71 |
| 100% | **106.28** | **16.10** | **44.26** | **14.30** |

## 7.3 QUALITATIVE RESULT

Fig. 4 presents two examples evaluated in TRAVEL platform. The first two rows show the UAV successfully following the instructions, maneuvering through the buildings, and ultimately locating a yellow vehicle. During this process, the UAV experienced camera view shifts due to its attitude

changes, highlighting the realistic fidelity of our platform. In contrast, the third row demonstrates a collision caused by insufficient altitude while navigating through a forested area, showcasing the challenges posed by complex environments. Additional results can be found in Appendix E.

## 8 CONCLUSION

We address the challenges of realistic UAV VLN from three aspects: platform, benchmark, and methodology. To achieve this, we develop the TRAVEL platform, which provides realistic environments, flight simulation, and comprehensive algorithmic support. We also construct a target-oriented realistic UAV VLN dataset and propose the UAV-Need-Help benchmark, which provides assistance to guide UAVs through complex VLN scenarios. Additionally, we propose a UAV navigation LLM along with a backtracking sampling-based data augmentation strategy, which together effectively enhance the performance of realistic trajectory-based VLN tasks. Our contributions establish a unified framework for realistic UAV VLN research, making significant strides toward bridging the gap between simulation and real-world UAV navigation applications. Furthermore, there are two promising directions for future research in realistic UAV VLN tasks. The first is to enhance the autonomous navigation capabilities of UAVs, enabling them to operate effectively in complex environments with minimal guidance. The second is to improve the transferability from UAV simulation to real-world deployment, facilitating the application of UAVs in real-world scenarios.

**Acknowledgements** This research is supported in part by National Key R&D Program of China (2022ZD0115502), National Natural Science Foundation of China (NO. 62461160308, U23B2010), "Pioneer" and "Leading Goose" R&D Program of Zhejiang (No. 2024C01161).

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

## A  PLATFORM AND DATASET DETAILS

We visually present some scenarios and complete object assets supported by the platform in Fig. 5 and Fig. 6, respectively. In addition to the scenarios shown in Fig. 5, we have also included 11 urban maps transferred from the CARLA simulator, and described the styles of different maps in Table 5. The dataset splits based on maps and objects are respectively presented in Table 6 and Table 7.

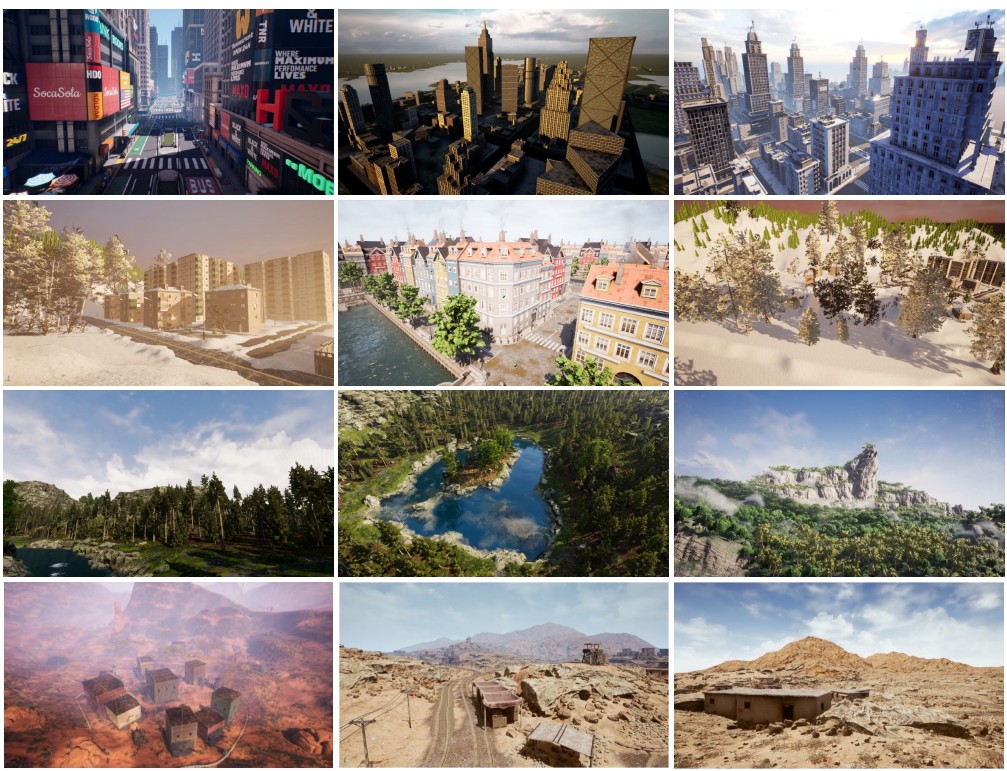

Figure 5: Platform Scenarios Overview: We present a selection of scenarios, respectively depicting city, town, forest, and desert environments.

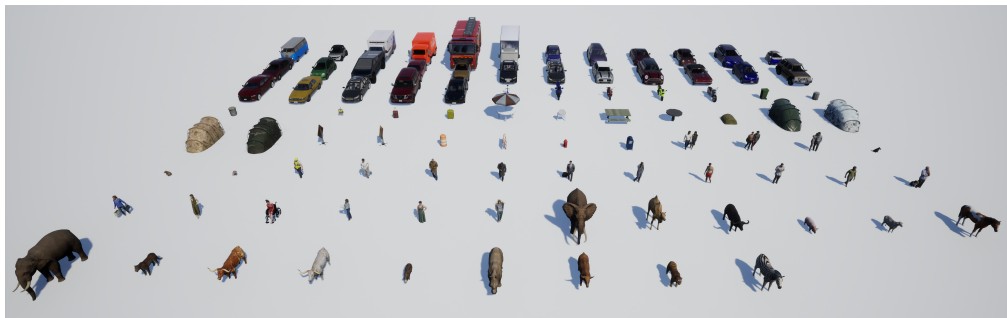

Figure 6: Platform Objects Overview: We present the platform's object assets, including vehicles, people, animals, and other objects.

## B  PROMPT ENGINEERING IN DESCRIPTION COLLECTION

As shown in Fig. 7, we use the following prompt to collect object descriptions (Section 4.1). The prompt integrates information from multiple camera views to provide a comprehensive representation of the target object's visual characteristics. The generated instruction incorporates detailed descriptions of the surrounding environment's colors and shapes to facilitate accurate visual identification. Furthermore, it specifies the spatial relationship between the target and nearby objects, enabling precise localization of the target within its context.

| Map | Style | Map | Style |
|---|---|---|---|
| NYCEnvironmentMegapa | City, Park, River | NewYorkCity | City |
| ModularPark | Village, Forest, Park | BrushifyForestPack | Forest |
| BattlefieldKitDesert | Desert | Carla_Town01 | Town, River |
| ModularEuropean | City, Park | Carla_Town02 | Town |
| TropicalIsland | Mountain, Forest | Carla_Town03 | City |
| BrushifyCountryRoads | Mountain, Village | Carla_Town04 | Town, Mountain |
| NordicHarbour | Town, River | Carla_Town05 | Town |
| Japanese_Street | City | Carla_Town06 | Highway |
| BrushifyUrban | City, River | Carla_Town07 | Village |
| London_Street | City | Carla_Town10HD | City |
| WesterTown | Desert, Village | Carla_Town15 | City, Forest |

Table 5: Maps and corresponding style classifications.

| Dataset | Maps |
|---|---|
| Train set | NYCEnvironmentMegapa, NewYorkCity, BattlefieldKitDesert, ModularEuropean, TropicalIsland, BrushifyCountryRoads, NordicHarbour, Japanese_Street, BrushifyUrban, London_Street, WesterTown, BrushifyForestPack, Carla_Town01, Carla_Town02, Carla_Town04, Carla_Town05, Carla_Town06, Carla_Town07, Carla_Town10HD, Carla_Town15 |
| Test seen | NYCEnvironmentMegapa, NewYorkCity, ModularEuropean, TropicalIsland, Carla_Town01, Carla_Town02, Carla_Town04, Carla_Town05, Carla_Town06, Carla_Town07, Carla_Town10HD, Carla_Town15 |
| Test unseen | Carla_Town03, ModularPark |

Table 6: Map-based dataset split.

| Dataset | Object |
|---|---|
| Train / Test seen | VolkswagenT2_2021_Parked, Tazzari, AmbulanceParked, CarlaCola, FiretruckParked, Truck, AudiA2, AudiTT, VolkswagenBeetle, BMWGrandTourer, BMWIsetta, ChevroletImpala, Citroen_C3, Cybertruck, Charger_parked, LincolnMkz2017_prop, MiniCooperS, Mini2021_parked, Mustang_prop, NissanPatrolST, ChargerParked, FordCrown_parked, LincolnParked, NissanPatrol2021_parked, TeslaM3_parked, RoadBike, Yamaha, Vespa, Kawasaki, Harley, Bin, TrashCan01, TrashCan03, TrasdhBag, ShoppingBag, ShoppingTrolley, TravelCase, PlasticTable, PlasticChair, Table, Table_Round, hayBale, Tent_02, Tent_03, Tent_04, WarningConstruction, WarningAccident, TrafficCones_01, StreetBarrier, FireHdrant, MailBox, Couple02, Couple04, Couple13, Dog01, Dog03, Man01, Man02, Man10, Man17, Man32, Man33, Man36, Man39, Man42, Man54, Woman04, Woman27, Woman44, Woman51, Woman52, Woman53, African_elephant, camel, Domestic_sheep, horse_skeleton, leopard, Longhorn_cattle, Longhorn_cattle_V2, mud_pig, rhino, Scotland_cattle, tiger, zebra |
| Test unseen | Man17, Man54, Woman51, Indian_elephant, Cape Buffalo, Domestic_pig, Dog02, Tent01, NissanMicra, EtronParked, SeatLeon, Lincoln2020Parked, TravelCase |

Table 7: Object-based dataset split.

## C EXPERIMENTAL DETAILS

**Implement Detail.** We include the implementation details necessary for reproducing results. In general, we adopted a training strategy similar to that described in Li et al. (2024). During the training of the MLLM, we freeze most of the model's parameters and only compute gradients on the visual projector, trajectory prediction head, and LoRA layers. These trainable parameters make up just 4% of the total model parameters and significantly reduce computational costs while maintaining the model's core capabilities. We supervise the predicted 3D angles using cosine similarity loss

**Prompt:**
**# CONTEXT #** Describe the object in the image and its surroundings. These are images captured from the top, front, back, left, and right views, and the center object has a mesh body name of {object name}.
**# OBJECTIVE #** Please relate information from different views to describe the visual characteristics of the central object and the surroundings as a whole, prioritize describing the surrounding objects > surrounding buildings > vegetation, and descriptions prioritize color and shape.
**# STYLE #** Describe objectively; do not use jargon such as grid body, etc.
**# TONE #** Do not use subjective evaluation words. For example: sunny, beautiful, etc.
**# AUDIENCE #** Describes the object and its surroundings to make it easier for the drone pilot to find the target.
**# RESPONSE #** Please write the above tasks together in a paragraph, keep it concise, just information, don't have any additional output, don't emphasize that there are multiple viewpoints, blend the information together, pay attention to differentiate whether it's a tall building or a short building, and what position the target is in.

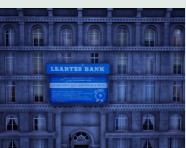 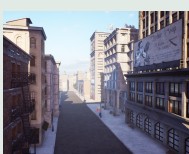 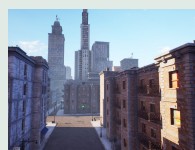 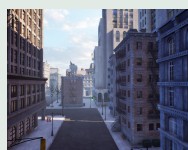 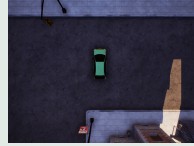

**Target direction:**
There is a target in front of UAV. Using your front as the x-axis and your right as the y-axis, The target is at a yaw angle of 213.0 degrees from you.

**Object description and environment information (GPT4):**
The central object is a green-colored vehicle positioned on a road. The vehicle is surrounded by urban architecture, including multiple tall and short buildings. The buildings are primarily multi-story, rectangular, and feature a mix of stone or brick facades. One of the notable buildings includes a "Leartes Bank" sign, indicating its purpose as a commercial building. There are a variety of structures, some with more ornate designs, particularly around windows and doors. Some of the buildings have fire escapes visible, typical of city settings. The streets are wide and largely empty, except for a few scattered objects like street lamps, signage, and the parked green vehicle. There is no visible vegetation or trees in the immediate area. The scene is devoid of people or other vehicles, adding to a sense of spaciousness. The road is straight and clean, with buildings positioned symmetrically on both sides. A billboard can also be seen on one of the buildings, contributing to the urban setting.

Figure 7: The instruction includes the creation of target direction, object descriptions, and environmental information. The target direction is determined based on the coordinates of the UAV's start and end positions, while the object descriptions and environmental information are generated by GPT-4 using the provided prompts, incorporating the top, front, rear, left, and right views and the object's name.

and apply L1 loss between the predicted waypoints and the ground truth. Similarly, for the trajectory completion model, we also freeze the parameters of the vision encoder. The MLP trajectory prediction layer is trained using L1 loss between the predicted trajectory and the labelled trajectory.

The MLLM model is trained on 8 NVIDIA A100 GPUs with a batch size of 128 for 2 epochs, while the fine-grained model and CMA model are trained with a batch size of 128 for 10 epochs. We use Adam optimizer with a one-cycle learning rate decay schedule to train all models, where we set the maximum learning rate to 5e-4.

Table 8: Ablation on model structure.

| Component | NE↓ | SR↑ | OSR↑ | SPL↑ |
|---|---|---|---|---|
| w/o LoRA | 124.28 | 12.20 | 36.52 | 10.76 |
| w/o $\mathcal{L}_{cos}$ | 113.53 | 13.55 | 40.28 | 12.34 |
| w/o $Q_{wp}$ | 111.53 | 15.18 | 41.13 | 13.61 |
| Ours | **106.28** | **16.10** | **44.26** | **14.30** |

**Ablation Study.** To validate the effectiveness of each part, we summarize experiments results in Table 8, where $L_{cos}$ indicates the cosine loss of poses, $Q_{wp}$ indicates the learnable query. The ablation of model architecture highlights the critical role of LoRA. The exclusion of LoRA resulted in an 8% decrease in OSR and 4% in SR. Removing the cosine loss and the learnable query also leads to a decline in performance, but the impact is relatively smaller. The complete model performs best across all metrics, validating the effectiveness of each component in enhancing model performance.

We further sample part of the test seen set for more extensive ablation experiments. First, we validate the effectiveness of the hierarchical prediction design. As shown in Fig. 8, the hierarchical trajectory prediction method achieves a balance between performance and computational efficiency, completing one inference in 0.407 seconds, which matches real-time drone navigation requirements. Using MLLM alone for long-distance prediction results in a decrease in success rate with similar inference times. Fine-grained predictions using MLLM improve performance by receiving assistance information more frequently. However, this comes at the cost of a significant increase in inference time, making it unsuitable for real-time navigation.

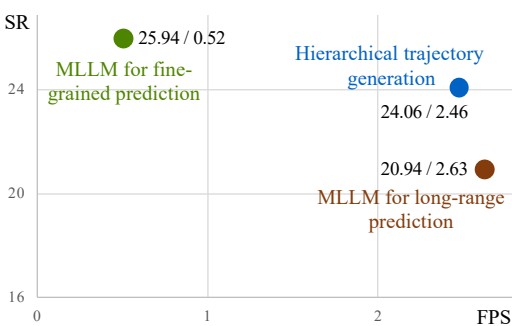

Figure 8: Ablation of hierarchical prediction.

Table 9: Ablation on assistant level.

| Assist Level | SR↑ | OSR↑ | NE↓ | SPL↑ |
|---|---|---|---|---|
| No assist | 6.25 | 11.88 | 167.10 | 4.95 |
| L3 assist | 7.20 | 13.71 | 163.55 | 6.17 |
| L1 assist | **24.06** | **43.13** | **130.57** | **20.42** |

Table 10: Ablation on sensor configurations.

| Model | SR↑ | OSR↑ | NE↓ | SPL↑ |
|---|---|---|---|---|
| Front&top view | 22.50 | 41.88 | 132.19 | 18.69 |
| Baseline | **24.06** | **43.13** | **130.57** | **20.42** |

We also conduct experiments on the UAV VLN task without relying on an assistant to validate the current capabilities of autonomous UAV navigation in Table 9. The results show that similar to previous UAV VLN tasks (Liu et al., 2023b), completing long-trajectory VLN tasks without guidance information remains a significant challenge.

In Table 10, we explore the impact of using fewer sensors. We believe the front and top views are more important for detection in search tasks, so we kept only these two perspectives for the VLN tasks. The experimental results show a slight performance drop across various metrics after reducing the number of sensors, indicating that visual information plays a supportive role in decision-making.

## D MLLM PROMPT TEMPLATE

In this section, we provide a detailed prompt for the input to MLLM, which organizes assistant instructions, drone states, task description information, and multi-view images into a specific format for input into MLLM.

```
A chat between a user and an AI agent.
The agent is a navigation model that outputs UAV waypoints based on
the user's instruction and UAV state information.

Assistant: <assist_info>
Previous displacement: <delta>
Current position: <cur_pos>
Current image: 
Instruction: <task_desc>

Please control the drone and find the target.
Next goal position: <waypoint>
```

Figure 9: We present the specific template for the input to MLLM, where <delta> represents the UAV's position change from the previous moment, <cur_pos> denotes the current global position in the coordinate system of the first frame, and <waypoint> is the learnable special query.

## E ADDITIONAL QUALITATIVE RESULTS

Here, we provide more visualization results of our method in closed-loop evaluation.

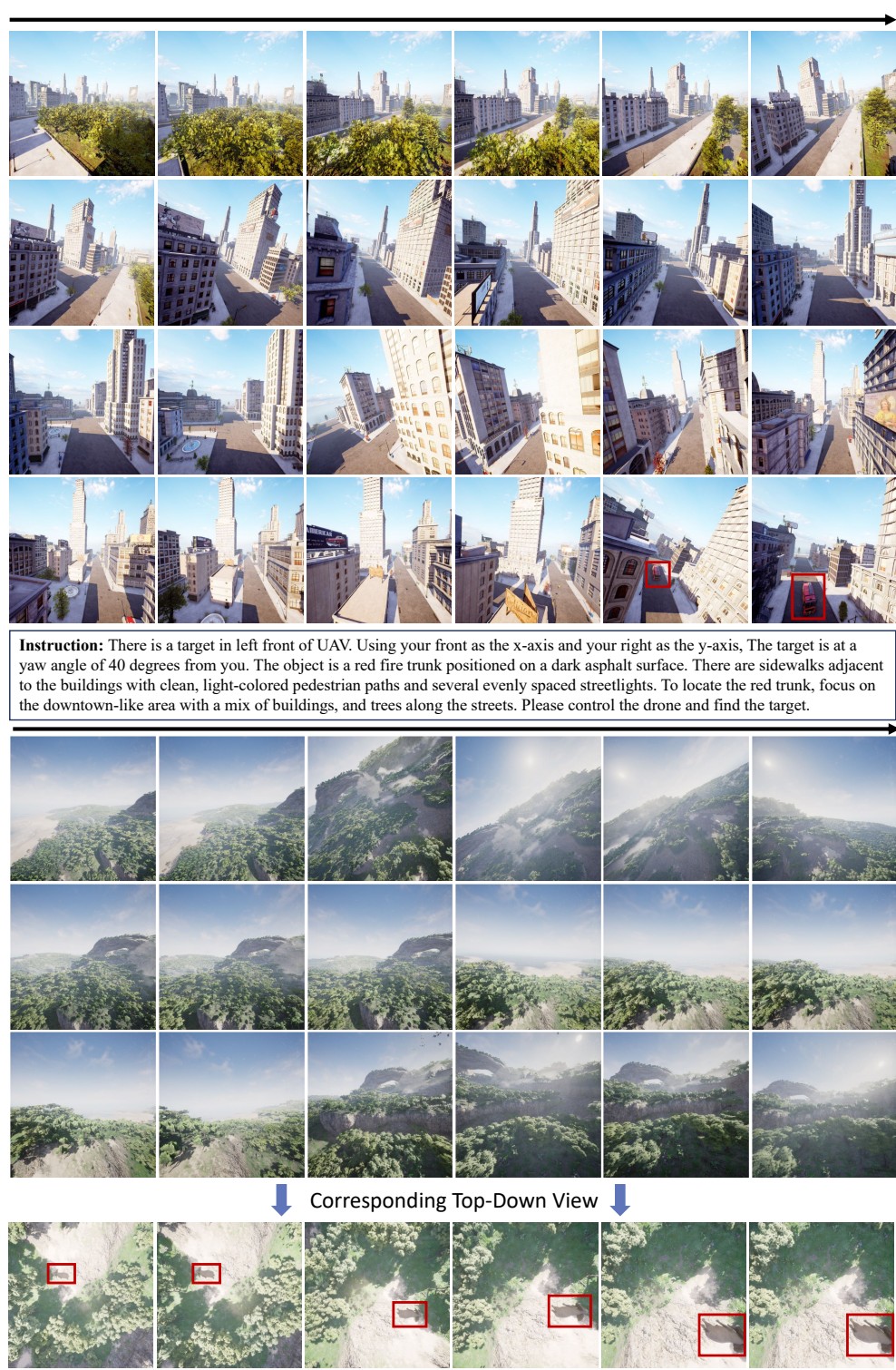

**Instruction:** There is a target in left front of UAV. Using your front as the x-axis and your right as the y-axis, The target is at a yaw angle of 40 degrees from you. The object is a red fire trunk positioned on a dark asphalt surface. There are sidewalks adjacent to the buildings with clean, light-colored pedestrian paths and several evenly spaced streetlights. To locate the red trunk, focus on the downtown-like area with a mix of buildings, and trees along the streets. Please control the drone and find the target.

Corresponding Top-Down View

**Instruction:** There is a target in the right front of UAV. Using your front as the x-axis and your right as the y-axis, The target is at a yaw angle of 30 degrees from you. The center object is a brown elephant lying on the ground surrounded by a dense area of green vegetation. To the front, back, left, and right, there are trees scattered throughout the landscape, with some areas displaying lighter of soil and rocks. The terrain is mostly flat with gentle slopes. In the backdrop, there are small hills and elevated areas visible under a clear sky, providing additional reference points for locating the horse. Please control the drone and find the target.

Figure 10: More examples of the UAV object search task using our method illustrate longer trajectories and improved navigation performance across various scenarios.

