# OpenReview forum: "Towards Realistic UAV Vision-Language Navigation: Platform, Benchmark, and Methodology"
_ICLR.cc/2025/Conference — ICLR 2025 Poster_

### Official Review · Reviewer_zwf5 · 2024-10-27

**Soundness:** 3
**Presentation:** 2
**Contribution:** 3
**Rating:** 6
**Confidence:** 2

**Summary:**

This paper introduces a UAV-based vision-and-language navigation benchmark built on the CARLA simulator. The simulator offers realistic environments and supports high-DoF control for UAV flight. In addition, the benchmark provides ground-truth human-controlled trajectories, trajectory descriptions summarized by a vision-language model (VLM), and assistive instructions for navigation evaluation. The paper also presents a vision-language model as a baseline for UAV-based vision-and-language navigation, achieving state-of-the-art performance on the proposed benchmark.

**Strengths:**

This paper is highly comprehensive, including a platform, benchmark and method technique for the VLN task in the domain of UAV. The authors have clearly significant efforts in completing this project.  This benchmark could help researchers investigate realistic UAV control under the task of VLN. Besides, the proposed LLM-based navigation framework shows strong performance among the baseline methods.

**Weaknesses:**

- This paper includes extensive content on the platform, benchmark, and method, but lacks many important details. For instance, `L398` mentions leveraging DAgger in a closed-loop simulation, yet no further experiments are provided to demonstrate its effectiveness.

- In my view, the L1 assistant (guiding the UAV to the oracle path) provides an oracle instruction by directly issuing ground truth commands, which should significantly enhance navigation performance. However, the experiment results show only a minor increase in SR (such as 16.1 in the full set).

- The visual encoding strategy does not account for historical information, as the navigation method relies solely on the latest frame `L373`, resulting in only 16+1 tokens for encoding visual data. This design neglects navigation history, raising the question of how navigation history can align effectively with the instructions.

- The paper lacks video results to illustrate the ground truth trajectory and the method's performance on this platform. Such visual demonstrations would provide a more intuitive understanding of this benchmark.

**Questions:**

- Why does the proposed method rely solely on the current observation? How does the model align visual history with the instructions? Additionally, the paper notes the use of `Llama-vid, Vicuna-7B, and EVA-CLIP`, similar to components used in another VLN paper [1]. A discussion comparing these approaches would be beneficial.

- In Tables 2 and 3, the OSR is significantly higher than the SR (almost double). Given the definition of OSR, methods can achieve high OSR by extensively exploring the environment. A discussion or additional experiments are needed to address this observation.

- Why not consider using existing LLM-based methods, such as [2], which employ waypoints in a manner similar to the proposed approach?

[1] Navid: Video-based VLM plans the next step for vision-and-language navigation

[2] NavGPT: Explicit reasoning in vision-and-language navigation with large language models

---

> ### Author Response · Authors · 2024-11-22
> **Response to Reviewer zwf5 (Part 1/2)**
>
> **Q1: Lack of experimental validation for DAgger**
>
> **A1:** Thank you for your valuable suggestion. We have taken the reviewers' feedback into account and have made necessary additions and revisions to the content of the paper.
>
> In fact, we have presented the model performance after incorporating DAgger in the original version of Table 2 (L447, "Ours-DA"). Compared to the base model, we observed improvements across various metrics, which demonstrates the effectiveness of integrating DAgger into our approach.
>
>
>
> **Q2:** **SR is not high even with L1 assistant**
>
> **A2:** Thank you for your insightful comment. We understand your concern about the relatively low SR despite the use of the L1 assistant. We believe the result of SR can be attributed to several factors:
>
> **Complexity of UAV VLN Tasks**: UAV-based VLN tasks are inherently complex. Previous works have reported relatively low success rates, such as AerialVLN[1] with an SR of 7.2% on its simple dataset and 2.9% on the full dataset, and CityNav[2] with an SR of 8.69%. Notably, AerialVLN uses step-by-step language instructions, and CityNav provides auxiliary information about landmarks near the endpoint on the map. Therefore, research on UAV-based VLN tasks in complex 3D environments is still in its early stages.
>
> **Understanding of Language Instructions**: Although the L1 assistant provides ground truth instructions, these commands still need to be interpreted by the model. For instance, the meaning of a "turn left" command depends on the context: it could be a slight turn if there is an obstacle on the right or a sharp 90-degree turn at an intersection. Thus, the model needs to correctly understand the language instructions. The performance of the Fixed baseline in Table 2 (L439) also shows that directly mapping instructions to a fixed trajectory does not yield ideal results. We provide visualizations of the fixed baseline trajectoriy and MLLM-predicted trajectory [here](https://anonymous.4open.science/r/anonymous_repo-6CCD/action_comparison.md).
>
> **Stricter Failure Settings**: Previous UAV tasks had insufficient collision detection. For example, AerialVLN places the UAV at a fixed position in the simulation, ignoring potential collisions during movement. CityNav focuses on high-altitude flight and does not consider interactions with objects. In contrast, we perform realistic UAV flights in the simulator and implement strict collision detection, accounting for obstacles such as power lines in urban environments and trees in villages and forests. This challenges both the model’s visual capabilities and the UAV's flight safety, which affects the overall task success rate.
>
>
>
> **Q3:** **Navigation history information**
>
> **A3:** We apologize for not clearly stating in the paper that our strategy was to add "historical displacement" and "global position" information in the coordinate system of the 0th frame in the textual input (we mentioned in the task setup that the UAV's state information can be accessed), allowing the UAV to understand changes in its own state over time.
>
> To validate this approach, we conducted ablation experiments on a small batch of data and analyzed performance across multiple dimensions, including success rate, inference time, and memory usage, for different configurations.
>
> | **Setting**                       | **SR** | **OSR** | NE     | SPL   | Time [s] | Memory [MB] |
> | --------------------------------- | ------ | ------- | ------ | ----- | -------- | ----------- |
> | Baseline (state info)             | 24.06  | 43.13   | 130.57 | 20.42 | 0.37     | 19757       |
> | 50 historical images              | 13.75  | 25.31   | 141.14 | 12.31 | 0.97     | 21432       |
> | 50 historical images + state info | 20.31  | 39.38   | 138.68 | 17.65 | 0.97     | 21443       |
>
> However, the results show that when only historical image information is used, the model's ability to understand historical data remains limited, resulting in some performance degradation. When both historical images and textual information are used, long sequences of multi-view images may interfere with the model's understanding, leading to slight performance drops. This might mean that the decision made based on the current frame is more crucial, and that historical visual information may play a limited role in complex, long trajectories.
>
> Moreover, in the dataset we collected, each trajectory consists of an average of 264 waypoints (as shown in Table 1, L119), meaning that extended steps are required to complete the task. Additionally, for object search tasks, the UAV needs multi-view images, with each frame in our setup containing 5 images. The large amount of image input results in increased inference time and higher memory usage, which also impacts the real-time nature of computation in UAV applications.

---

> ### Author Response · Authors · 2024-11-22
> **Response to Reviewer zwf5 (Part 2/2)**
>
> **Q4:** **Lack of video results for visual demonstration**
>
> **A4:** Thank you for your valuable suggestion. To provide a more intuitive understanding of the benchmark, we have provided visual comparisons between the ground truth trajectory and the model's predicted trajectory. Additionally, we have included first-person perspective videos for further comparison at this [link](https://anonymous.4open.science/r/anonymous_repo-6CCD/flight_visualization.md).
>
>
>
> **Q5:** **Reliance on current observation and comparison with related methods**
>
> **A5:** Thank you for your insightful question. As mentioned in A3, to address this challenge, we incorporate the UAV’s "historical displacement" and "global position" information into the text input to the large model, allowing the UAV to maintain an awareness of the goal's orientation specified in the instructions. We have validated the effectiveness of this design through ablation experiments.
>
> Regarding the use of Llama-vid, EVA-CLIP, and Vicuna-7B, Llama-vid is an excellent pre-trained multimodal model, and we have built our approach on top of it. Therefore, the image encoder (EVA-CLIP) and the large language model (Vicuna-7B) align with Llama-vid in terms of their architecture.
>
> Unlike Navid, which uses long image sequences for grounded-based vision-language navigation, we supplement our model with text-based UAV state information and introduce a learnable special token <waypoint>, that interacts with the visual-linguistic information for feature extraction.
>
> In terms of model output, Llama-vid generates dialogue text, and Navid outputs action types in text form. In contrast, our model uses the features extracted by the <waypoint> token to predict the next trajectory point for the UAV.
>
>
>
> **Q6:** **OSR significantly higher than SR**
>
> **A6:** The discrepancy where OSR is significantly higher than SR is indeed an issue worth noting. This phenomenon is generally caused by difficulties in endpoint localization, leading to the inability to terminate the task.
>
> In UAV-based VLN tasks, the scene space is large, and the agent has a high degree of freedom in flight. However, some target endpoints are small and difficult to detect accurately, such as pedestrians in urban environments or small animals in forested scenes. These challenges make it difficult to accurately localize the endpoint, which results in OSR being higher than SR. This phenomenon is also observed in existing UAV-based VLN works. For example, in the AerialVLN task, the SR is 2.9%, while the OSR is 10.2%; in the CityNav task, the SR is 8.69%, while the OSR is 35.51%. These studies also demonstrate a similar discrepancy between OSR and SR.
>
>
>
> **Q7:** **Existing LLM-Based method (NavGPT)**
>
> **A7:** Although NavGPT is indeed a valuable work in the field of indoor navigation, it differs fundamentally from our approach. In NavGPT, the action space of the agent is limited to a set of predefined discrete viewpoints that form a navigation graph, and the agent can only move between these viewpoints. This method corresponds to a discrete VLN task, as clearly stated in their paper: "the agents' action space is confined to the navigation graph G." In contrast, our goal is to directly output arbitrary points in a continuous space, rather than discrete actions or fixed locations, making it a task of realistic continuous trajectory VLN. Therefore, NavGPT does not align with our task setup.
>
>
>
> **References**:
>
> [1] AerialVLN: Vision-and-Language Navigation for UAVs
>
> [2] CityNav: Language-Goal Aerial Navigation Dataset with Geographic Information

---

> > ### Comment · Reviewer_zwf5 · 2024-11-23
> > **Response to Authors**
> >
> > Thanks to the authors' responses, some of my concerns have been resolved.
> >
> > However, I'm still confused about the navigation history information. I didn't find the so-called terms "historical displacement" and "global position" information in the paper. Does this mean you added these textual state descriptions to the input? There should be a more clear explanation because they are one of the key inputs to the model.
> >
> > Besides, leveraging state information does not answer my question regarding why the absence of historical visual information can align the task instruction with the navigation history.  For example, when there are repeated landmarks, how does the model leverage state information to distinguish whether it is the first encounter or the second time?

---

> > > ### Author Response · Authors · 2024-11-23
> > > **Response to Reviewer zwf5**
> > >
> > > Thanks a lot for your reply. First, we completely agree with your suggestion to explain the model's input more clearly. Our previous revision only mentioned incorporating state information into the text. Based on your suggestion, we have added the specific format of the prompt used in the MLLM model in the appendix E.
> > >
> > > Additionally, there might be some misunderstanding regarding the consistency between task instructions and navigation history. Our UAV VLN task is target-oriented and assistant-guided. The instruction only contains descriptions of the target direction, target object and surrounding environment, along with assistance guidance. Unlike the instructions in AerialVLN[1], which contain visually-grounded landmarks and require alignment with extensive visual history, our instructions do not include intermediate landmarks from the trajectory. Moreover, our proposed assistance mechanism encourages the model to focus more on the current visual and state information to make decisions.
> > >
> > > We would also like to clarify the connection between the state info and task instructions. The historical displacement refers to the change in the drone's position from the previous moment, helping the model understand the assistant's instructions for adjusting the subsequent planning. The current global position allows the drone to perceive its orientation relative to the starting position, aligning with the target direction in the task instructions. This assists the drone in completing the task instructions.
> > >
> > > **References:**
> > >
> > > [1] AerialVLN: Vision-and-Language Navigation for UAVs

---

> > > > ### Comment · Reviewer_zwf5 · 2024-11-23
> > > > **Response to Authors**
> > > >
> > > > Thanks for the quick responses. It now makes sense to me if the task is 'target-oriented and assistant-guided.' However, I am uncertain whether this design is also applicable to vision-and-language navigation, or if it is more aligned with target searching or following. Since I am not an expert in UAVs, I will rely on the consensus of other reviewers for this definition.

---

> > > > > ### Author Response · Authors · 2024-11-23
> > > > > **Response to Reviewer zwf5**
> > > > >
> > > > > Thank you very much for your understanding and quick response. Regarding your concern about whether the target-oriented and assistant-guided design can be applied to VLN tasks, there are several survey papers on VLN [1, 2] that discuss the classification of VLN tasks, which can be referenced. They typically categorize tasks based on target descriptions as **Coarse-grained** or **Goal-Oriented** VLN tasks, while tasks with additional guidance are referred to as **Oracle Guidance** or **Passive, Multi-turn** VLN tasks. Therefore, this VLN task design could be considered reasonable.
> > > > >
> > > > > **References:**
> > > > >
> > > > > [1] Vision-and-Language Navigation:  A Survey of Tasks, Methods, and Future Directions
> > > > >
> > > > > [2] Vision-Language Navigation: A Survey and Taxonomy

---

### Official Review · Reviewer_7iwn · 2024-11-03

**Soundness:** 3
**Presentation:** 2
**Contribution:** 2
**Rating:** 6
**Confidence:** 3

**Summary:**

The paper proposes a new UAV platform with several integrated scenes, continous action space and 6 DoF instead of previously 4. In addition they provide a VLN dataset together with a benchmark “UAV-Need-Help” together with an assistant that can provide guidance in three different levels. Lastly they have a model based on a LLM with a hirarchical strucutre that first predicts the target pose and then predicts the fine-grained path trajectory. The authors evaluate their model closed-loop and show qualitative results of the language predictions of the LLM.

**Strengths:**

S.1. Complete set to research UAVs. The paper provides a platform/ simulation environment, dataset, benchmark and a baseline model. This opens up a lot of possibilities for the community to do research in this field.

S.2. Baselines: Having multiple baselines for a new benchmark is important. I also appreciate the human baseline.

S.3. Hierarchical prediction: The concept of hierarchical predictions is interesting and makes sense. (But missing ablation to show that hierarchical is superior to directly predicting the fine-grained trajectory).

**Weaknesses:**

**W.1. Comparison to existing work**

1. The works seems to be quite similar to [AerialVLN](https://arxiv.org/abs/2308.06735) or CityNav. One difference mentioned is the DoF and the use of discrete actions. The authors of AerialVLN state “Although the simulator **supports flying towards any given direction and speed/distance**, we consider the eight most common low-level actions in UAVs:Move Forward, Turn Left, Turn Right, Ascend, Descend, Move Left, Move Right and Stop.”, which means in my understanding that they do support continuous actions. There dataset also includes language instructions. So I was wondering why this new simulator and dataset is needed and can’t be build on top of the existing one? It would be great if the authors could highlight unique features or improvements their simulator offers, and explain why building on existing platforms was not sufficient for their research goals.
2. Difference to AirSim: AirSim does provide an API and a human control interface supporting PX4. In the paper it sounds like this is a contribution of the new platform OpenUAV (e.g. line 212 ff.). It would be good to make it more clear in the whole paper what is used from existing works and which parts are new in this paper.
3. Difference to related work that combines UE + AirSIM: The authors state that several works “have combined UE with AirSim to develop high-fidelity imaging platforms. However, these platforms either suffer from low rendering precision or lack algorithmic support for continuous trajectory VLN tasks”. (line 149) Since the authors also combine UE with AirSim, it would be great if the differences to the existing works could be described in more detail.

**W.2. Details of descriptions/ Clarity**

1. Naming: There is already an OpenUAV platform (see[1]). It would be good to use an unique naming and also elaborate the differences.
2. Scenarios: It would be great to have more details of the available scenarios. For me it is not clear what a scenario means in this context? Is it a different environment (e.g. forest vs. city) or is it similar to CARLA a specific event happening (see https://leaderboard.carla.org/scenarios/)? What are the 22 distinct scenarios available in this simulator? If the authors could provide a list or table of the 22 scenarios, including a brief description of each it would help a lot.
3. Dataset splits: Figure 2 is great to get a rough overview of the objects but a detailed overview in the supplementary with all available objects and scenes and which are used for which split would be helpful.
4. Figure 3 (minor): In the image it seems that the front view image is input to the Trajectory completion module but in the text the authors say “it takes the front-view visual token T_img” (line 386). It would be good to have this clearer in the figure.
5. DAgger: What exactly is the teacher model? Can you provide more details?

**W.3. Evaluation/ Benchmark**

1. As I understood for all results an oracle path is necessary to generate the assistants. An additional mode without any assistants would be good to judge how hard the task is and in an ultimate fully autonomous setting this would be the wished setting I assume. The authors state “the inherent complexity and dynamic nature of aerial environments, with obstructed views and shifting perspectives, make basic target descriptions insufficient for UAV” (line 323). I am wondering if temporal input and a memory could help for this making the assistants unnecessary?
2. The authors say “UAV trajectories are inherently continuous and difficult to decompose into discrete actions, leading to unrealistic navigation when such simplifications are applied” (line 081). They use (different to previous works) continuous action space. To validate this statement it would be good to have a discrete baseline and qualitative example of the shortcomings (which are solved by the continuous version).

[1] H. Anand *et al*., "OpenUAV Cloud Testbed: a Collaborative Design Studio for Field Robotics," *2021 IEEE 17th International Conference on Automation Science and Engineering (CASE)*

**Questions:**

I am not an expert in the field of UAV but after looking at the related work it is still a bit unclear for me what the main contributions are and how it differentiates to existing work. It would be good if the authors could make it clearer what already exists and what is new. Secondly it would be really helpful to include more ablations on the importance of continuous actions or hierarchical predictions.

Minor suggestions/ questions:
1. Make sure that the abbreviations are introduced properly. E.g. line 016 is the first time you mention UAV, it would be good to have the long version when first using the short version.
2. Human baseline: I would be curious how well a human performs in the L3 setting. Is the human baseline with a person who tried it several times and is trained or is it a “zero-shot” baseline?

---

> ### Author Response · Authors · 2024-11-22
> **Response to Reviewer 7iwn (Part 1/4)**
>
> **Q1:** **Similarities with AerialVLN/CityNav and the need for a new platform and dataset**
>
> **A1:** Thank you for your insightful question.
>
> While AerialVLN[1] and CityNav[2] are valuable benchmarks, they focus on different aspects of the problem. **AerialVLN** primarily focuses on sentence-by-sentence language instruction navigation in urban environments and uses discrete action sequences. In its specific implementation, they ignore the drone's flight process and directly set the drone at a specified position based on the model's current decision. This approach: 1) overlooks potential collisions during motion (e.g., a single action displaces the drone by 5 meters), and 2) keeps the drone always horizontally oriented, with only the yaw angle changing, while the roll and pitch remain unchanged. **CityNav** also uses a city-level simulator for goal-directed language instruction navigation with discrete action sequences, but it sets the drone's flight altitude very high (100-150 meters), resulting in a perspective more similar to remote sensing imagery, and the simulator does not handle interactions with objects (such as collisions).
>
> In contrast, we focus on target-oriented language instruction navigation in urban and natural environments, where the flight altitude in the reference trajectories is relatively low. In urban scenes, the drone needs to navigate between buildings, with strict collision detection implemented. Additionally, we use waypoint-level control granularity to guide the UAV in continuous flight, capturing sensor data based on realistic UAV attitudes to complete the VLN task.
>
> In terms of the platform, AerialVLN achieves excellent simulation results using the combination of UE and AirSim, but there are several aspects that still need to be expanded:
>
> **Scene and object assets**: AerialVLN open-sourced the precompiled environment for Linux, which limits secondary development, especially the ability to add object assets. Many target-related tasks, such as search and perception, cannot be conducted. In contrast, we aim for the platform to support a broader range of drone tasks, which is why we plan to open-source the project source code and compiled versions of various scene files. Additionally, our data collection framework includes a user-friendly API for placing target objects in scenes.
>
> **Continuous and realistic trajectory data**: AerialVLN currently adjusts the UAV's position by "directly setting its posture"， which lacks the changes in attitude angles during flight. In contrast, we have developed a framework for continuous trajectory collection based on time intervals, inspired by the OpenCDA[3] in the autonomous driving field. This framework captures poses at equal time intervals rather than fixed action sequences, significantly reducing the domain gap between simulated and real UAV flight trajectories.
>
> We provide a comparison of the trajectories between AerialVLN and our dataset in an anonymous [link](https://anonymous.4open.science/r/anonymous_repo-6CCD/trajectory_visualization.md) to offer a more intuitive understanding of the task differences.
>
>
>
> **Q2:** **Difference from AirSim and clarification of existing vs. new function**
>
> **A2:** Thank you for pointing out the unclear contribution description. We will clarify this in the paper. As you mentioned, AirSim is indeed a powerful drone control plugin, and we used its core functionalities to build our platform. However, the control link using the remote controller-QGC-PX4-simulation environment that AirSim supports is relatively complex, especially when it comes to tasks like environment setup and remote controller calibration.
>
> To address this, we extended the existing functionality by developing user-friendly APIs. This feature allows users to simulate remote controller input via the keyboard, and we have integrated it into our data collection framework.
>
>
>
> **Q3:** **Issues in describing related work and differences with UE + AirSim platforms**
>
> **A3:** We sincerely apologize for the confusion caused during your reading. We intended to highlight that widely used simulation platforms like Gazebo[4], which are commonly employed for UAV control, suffer from limitations in visual fidelity and rendering accuracy. Unfortunately, this point was not clearly conveyed in the original version of the paper. We have now updated and clarified the description in the revised version for better clarity.
>
> Regarding the differences with existing works that combine UE and AirSim, our focus lies primarily on the platform's task-specific capabilities and algorithmic support. As mentioned in A1, we aim to address specific challenges in continuous realistic trajectory-based VLN tasks, where previous platforms may have lacked support for such tasks.

---

> ### Author Response · Authors · 2024-11-22
> **Response to Reviewer 7iwn (Part 2/4)**
>
> **Q4:** **OpenUAV platform naming**
>
> **A4:** Thank you for your valuable feedback regarding the naming of our platform. To address this concern, we have decided to rename our simulation platform to **TRAVEL**, which stands for **T**owards **R**ealistic **A**erial **V**ision and **E**xploratory **L**anguage Navigation. This new name better reflects the unique focus and goals of our platform.
>
>
>
> **Q5:** **Clarification of scenario**
>
> **A5:** Thank you for your suggestion. In the paper, "scenarios" refer to different environments or maps available within the platform. Currently, the platform supports 22 distinct scenarios, and we provide descriptions of each scenario in this [link](https://anonymous.4open.science/r/anonymous_repo-6CCD/scenario&asset_details.md).
>
>
>
> **Q6:** **Dataset splits details**
>
> **A6:** Thank you for your valuable suggestion. We have provided visual images of objects, as well as the split infomation of scenes and objects in this [link](https://anonymous.4open.science/r/anonymous_repo-6CCD/scenario&asset_details.md).
>
>
>
> **Q7:** **Clarification of input to Trajectory Completion module**
>
> **A7:** Thank you very much for pointing that out. The description of the input to the Trajectory Completion module in the paper was indeed unclear. Our intention was to indicate that the Trajectory Completion module takes the front-view image as input, where the front-view RGB image is encoded by ViT to generate the front-view token $T_{img}^f$. We have revised the paper to clarify this point.
>
>
>
> **Q8:** **Confusion on the teacher model in DAgger**
>
> **A8:** Thank you for your question. The teacher model is an expert policy that guides the UAV during the DAgger process. Specifically, in our DAgger implementation, the teacher model directly receives the closest position to the current location from the ground truth (GT) trajectory and computes a trajectory sequence, which serves as the UAV's path for the next time step. We have revised the paper to clarify this point.

---

> ### Author Response · Authors · 2024-11-22
> **Response to Reviewer 7iwn (Part 3/4)**
>
> **Q9:** **UAV autonomous setting**
>
> **A9:** Currently, research on UAV-based VLN tasks is still limited, and significant **challenges** remain in achieving visual-language understanding and long-term 3D spatial planning in dynamic environments. In designing our assistants, we were primarily motivated by the low success rates of previous UAV VLN methods. For instance, AirealVLN achieved a success rate (SR) of only 7.2% on its simple dataset and 2.9% on the full dataset, while CityNav had an SR of 8.69%, even when assistance was provided by indicating landmarks near the destination on the map. Additionally, our task imposes higher demands, such as more realistic trajectory generation and stricter collision avoidance. As a result, we developed assistants at multiple levels. We envision the highest level of assistance to be similar to a navigation system used when driving in an unfamiliar environment, while lower levels provide more lightweight prompts to gradually guide the UAV toward autonomo us planning.
>
> It is worth noting that the L3 assistant does not require an oracle path to generate its instructions. Instead, it relies solely on depth map information to issue steering commands when the UAV faces imminent danger (e.g., a potential collision), guiding the UAV to avoid obstacles. The decision of how to reach the target remains autonomously made by the UAV. As shown in the results of Table 2, the success rate with L3-level assistance is relatively low, highlighting the challenges of the task. We have added a performance comparison without any assistance using a small batch of test data. The experiment shows completing long-trajectory VLN tasks without guidance still presents challenges.
>
> | **Assist Level** | **SR** | **OSR** | NE     | SPL   |
> | ---------------- | ------ | ------- | ------ | ----- |
> | No assist        | 6.25   | 11.88   | 167.10 | 4.95  |
> | L3 assist        | 7.20   | 13.71   | 163.55 | 6.17  |
> | L1 assist        | 24.06  | 43.13   | 130.57 | 20.42 |
>
> We have also explored the impact of temporal input and memory mechanisms on model performance. In our dataset, the average trajectory contains 264 points, and search tasks require multi-view images. When using historical images, the visual input to the large model becomes quite extensive. Therefore, we only added the current position and historical displacement information to the input of the large model, while focusing on the current visual information for decision-making. We conducted ablation experiments using historical image information, as shown in the table below.
>
> | **Setting**                       | **SR** | **OSR** | NE     | SPL   | Time [s] | Memory [MB] |
> | --------------------------------- | ------ | ------- | ------ | ----- | -------- | ----------- |
> | Baseline (state info)             | 24.06  | 43.13   | 130.57 | 20.42 | 0.37     | 19757       |
> | 50 historical images              | 13.75  | 25.31   | 141.14 | 12.31 | 0.97     | 21432       |
> | 50 historical images + state info | 20.31  | 39.38   | 138.68 | 17.65 | 0.97     | 21443       |
>
> The experiment shows that when only historical image information is used, the model's ability to understand historical data remains limited, resulting in some performance degradation. When both historical images and textual information are used, long sequences of multi-view images may interfere with the model's understanding, leading to slight performance drops. On the other hand, the growth of image sequences increases inference time and memory usage, which also impacts the real-time nature of computation.
>
>
>
> **Q10:** **Discrete baseline and qualitative example**
>
> **A10:** Thank you for your suggestion. We have defined a "Fixed" baseline in the Comparison Baselines section (line 420) and provided the corresponding evaluation results in Table 2. As shown, there is a clear difference between the fixed baseline and the MLLM model using continuous trajectory prediction. We provide visualizations of the fixed baseline trajectory and MLLM-predicted trajectory [here](https://anonymous.4open.science/r/anonymous_repo-6CCD/action_comparison.md), along with a detailed analysis to illustrate the shortcomings addressed by the continuous action space. Additionally, there may have been some misunderstanding. The continuous, realistic trajectories are part of our task setup. Therefore, even in the fixed action setting, as shown in our visualizations, the predicted points are still flown by the UAV along realistic trajectories.

---

> ### Author Response · Authors · 2024-11-22
> **Response to Reviewer 7iwn (Part 4/4)**
>
> **Q11:** **Clarification of main contributions and ablation studies on continuous actions or hierarchical predictions.**
>
> **A11:** Thank you for your feedback. We understand your confusion and would like to clarify the differences between our contributions and existing work:
>
> **Platform** – Existing UAV VLN platforms often lack APIs that control the continuous and realistic flight dynamics of UAVs, and their extensibility is limited. To address this, we have designed a platform that supports realistic, continuous UAV trajectories. Our platform includes a rich set of expandable scenes and objects, offering features for object placement, UAV control, data collection, and closed-loop simulation.
>
> **Dataset** – Existing UAV VLN datasets consist of discrete action sequences, which differ significantly from the data collected during real UAV flights in actual environments. Therefore, we have collected 12k trajectories and instruction data, sampled at equal time intervals and at the waypoint granularity.
>
> **Benchmark** – Existing UAV VLN benchmarks are designed around discrete action sequences, and their challenging task settings often result in low success rate. To address this, we implemented multi-level assistant mechanisms and obtained real-time sensor data from continuous UAV flights with real poses.
>
> **Model** – Most UAV VLN models rely on traditional VLN baseline models, while some ground-based VLN tasks have begun utilizing large models for high-level decision-making or discrete control outputs. In contrast, we propose a framework that combines both small and large models to collaboratively plan UAV trajectory sequences with hierarchical outputs.
>
> We answered the question regarding continuous action space in A10. We further conducted an ablation analysis on the hierarchical trajectory generation model in terms of performance and computational efficiency. The table below shows that using MLLM alone for long-distance prediction results in a decrease in success rate with similar inference times. Fine-grained predictions using MLLM improve performance by receiving assistance information more frequently. However, this comes at the cost of a significant increase in inference time, making it unsuitable for real-time navigation.
>
> | **Model**                          | SR    | OSR   | NE     | SPL   | Time [s]     |
> | :--------------------------------- | ----- | ----- | ------ | ----- | ------------ |
> | Hierarchical trajectory generation | 24.06 | 43.13 | 130.57 | 20.42 | 0.38 + 0.027 |
> | MLLM for long-range predictions    | 20.94 | 40.00 | 137.58 | 17.99 | 0.38         |
> | MLLM for fine-grained predictions  | 25.94 | 50.31 | 126.24 | 22.67 | 1.90         |
>
>
>
> **Q12:** **Abbreviation of UAV**
>
> **A12:** We noticed the abbreviation issue with UAV. Thank you for pointing this out. We will make corrections in the upcoming version update.
>
>
>
> **Q13:** **Human baseline**
>
> **A13:** Our human baseline was tested after multiple attempts and training. Under L3, even humans need sufficient exploration to find the target described by the task. This indicates that the drone navigation model must have strong environmental awareness and autonomous exploration capabilities to complete the task.
>
>
>
> **Reference:**
>
> [1] AerialVLN: Vision-and-Language Navigation for UAVs
>
> [2] CityNav: Language-Goal Aerial Navigation Dataset with Geographic Information
>
> [3] OpenCDA: An Open Cooperative Driving Automation Framework Integrated with Co-Simulation
>
> [4] Design and use paradigms for Gazebo, an open-source multi-robot simulator

---

> > ### Comment · Reviewer_7iwn · 2024-11-27
> >
> > Thanks a lot for the detailed response. I highly appreciate the time and effort.
> > I keep my positive score and recommend acceptance.
> >
> > However, I have two follow up points.
> > First, I think it would be very useful if the information about which objects and maps are unseen in the unseen splits would be added to the supplementary (currently provided in anonymous link).
> >
> > And Second, from the unseen object split I was wondering if it is fair to claim good zero-shot generalisation since for most objects very similar objects are in the train set. For example the African Elefant is in the train split and the Indian Elefant is in the test split. Wouldn’t it be better to have a unseen object split where whole categories of objects are unseen (e.g all elefants, all dogs and so on) to properly evaluate generalization to unseen objects?
> > I don’t expect those experiments and changes to be done during the rebuttal but I think for the future it would be very valuable to have a proper generalisation split.

---

> > > ### Author Response · Authors · 2024-11-27
> > > **Response to Reviewer 7iwn**
> > >
> > > Dear Reviewer 7iwn,
> > >
> > > Thank you very much for your feedback. We are glad to have addressed some of your previous concerns. Regarding the two follow-up points you raised, we take them very seriously and plan to make further improvements based on your suggestions.
> > >
> > > First, we will add more detailed information about the objects and scenes in the supplementary material to help readers gain a clearer understanding of the dataset.
> > >
> > > Second, thank you for your valuable feedback on the dataset splits! Currently, our dataset splits are based on sampling from the overall categories, and we split objects using features such as attire for people or color for animals. As you pointed out, there are still some examples that are visually similar. Your suggestion is very insightful, and adopting the "completely unseen object categories" split (for example, all elephants, all dogs, etc.) would indeed be a more effective way to evaluate the model's generalization ability. We fully agree with your point and plan to update our dataset to provide a more valuable test of generalization capability.
> > >
> > > Once again, thank you for your constructive suggestions. Your feedback is crucial to the improvement of our work!
> > >
> > >
> > > Best regards,
> > >
> > > Authors of Paper 611

---

> ### Author Response · Authors · 2024-11-26
> **Response to Reviewer 7iwn**
>
> Dear Reviewer 7iwn,
>
> Thank you for your valuable suggestions. In response to your comments and concerns raised in the previous round of review, we have provided further clarifications and revisions in the following areas: Comparison to Existing Work, Details of Descriptions / Clarity, Evaluation / Benchmark, Clarification of Main Contributions, and Ablation Studies.
>
> We would greatly appreciate it if you could consider our additional explanations and revisions.
>
> Thank you again for your time and thoughtful review.
>
> Best regards,
>
> Authors of Paper 611

---

### Official Review · Reviewer_LL2n · 2024-11-04

**Soundness:** 3
**Presentation:** 3
**Contribution:** 3
**Rating:** 6
**Confidence:** 4

**Summary:**

This paper proposes a new framework for UAV navigation, focusing on the different challenges of aerial environments and ground-based agent navigation.
A realistic UAV simulation platform, OpenUAV, is proposed, which supports continuous six-degree-of-freedom (6-DoF) trajectories, realistic flight dynamics, and diverse environmental scenarios. The platform integrates the AirSim plugin and Unreal Engine 4 (UE4) to achieve high-fidelity simulation.

UAV-Need-Help Benchmark: A new benchmark task introduces an assistant-guided UAV target search task, where the assistant provides different degrees of navigation guidance to the UAV to help it cope with complex navigation tasks.

UAV Navigation Large Model (LLM): A multimodal model is proposed that can process visual and textual information simultaneously and generate hierarchical navigation trajectories to address the complexity of UAV navigation in three-dimensional space. The platform, dataset, and method aim to narrow the gap between simulated UAV navigation tasks and practical applications, and provide more realistic and challenging tasks for UAV VLN research.

**Strengths:**

1. Comprehensive data set: The author contributes a large-scale goal-oriented UAV VLN data set, including 6-degree-of-freedom flight trajectories, providing a large number of training and evaluation resources for the research community.
2. Real UAV simulation: The OpenUAV platform significantly improves the authenticity of UAV navigation tasks, supports continuous flight trajectories, diverse environments and realistic flight dynamics, and solves the limitations of previous UAV VLN research.
3. Innovative benchmark task: The UAV-Need-Help benchmark introduces a challenge of practical significance, through different levels of help mechanisms, to cope with the challenges caused by the inherent dynamics of the aerial environment, obstructed views and changes in perspective.
4. Hierarchical trajectory generation: The proposed UAV navigation large model (LLM) combines long-range trajectory prediction and fine-grained control, and can handle the hierarchical complexity in UAV navigation tasks.
5. Parallelization and scalability: The platform supports efficient data collection and evaluation in parallel environments, which is crucial for large-scale UAV research.

**Weaknesses:**

Please see the Questions section for detailed improvement suggestions and questions.

**Questions:**

1. Real-world Deployment:
How well do the models and methods developed on the OpenUAV platform perform in real-world UAV navigation tasks? Has there been any experimental validation involving physical UAVs? Although the simulation platform is quite realistic, it remains unclear to what extent the models trained on this platform can generalize to real-world UAV tasks. A comparison with experiments using real UAVs is lacking.

2. Autonomy and Reliance on Assistance:
The UAV-Need-Help benchmark seems overly reliant on the assistance mechanism, which could reduce the autonomy of the UAV. It is unclear how the UAV Navigation LLM would perform without such assistance. Can the UAV autonomously recover from navigation failures in the absence of external help, particularly when faced with obstacles?

3. Efficiency and Real-time Performance:
Considering the complexity of the LLM-based navigation model, how feasible is the real-time deployment of such models on resource-constrained UAVs? Is there any analysis of the computational requirements and latency when running these models on actual drones?

4. Ablation Studies:
Could more insights be provided on the contribution of individual components (e.g., assistant guidance, hierarchical trajectory generation, sensor configurations)? How would the model's performance change if fewer sensors were used, or if the advanced trajectory decoder based on MLLMs were removed?

5. Lack of Comparison with Ground-based VLN:
While the paper highlights the differences between ground-based and UAV navigation, there is a lack of direct comparison with existing ground-based VLN methods(Seq2Seq[1], DUET[2], NavGPT2[3]). It is unclear what unique challenges the proposed framework addresses. Specifically, what are the differences between ground-based VLN methods and the UAV Navigation LLM? Could ground-based VLN methods be applied to OpenUAV, and if so, how would they perform?

6. Extensibility to Other Tasks
Besides the object search task, can the proposed platform and dataset be generalized to other UAV tasks such as mapping, exploration, or surveillance? Would the dataset require significant modifications to accommodate these tasks?

[1]Anderson, Peter, et al. "Vision-and-language navigation: Interpreting visually-grounded navigation instructions in real environments." Proceedings of the IEEE conference on computer vision and pattern recognition. 2018.
[2]Chen, Shizhe, et al. "Think global, act local: Dual-scale graph transformer for vision-and-language navigation." Proceedings of the IEEE/CVF Conference on Computer Vision and Pattern Recognition. 2022.
[3]Zhou, Gengze, et al. "Navgpt-2: Unleashing navigational reasoning capability for large vision-language models." European Conference on Computer Vision. Springer, Cham, 2025.

---

> ### Author Response · Authors · 2024-11-22
> **Response to Reviewer LL2n (Part 1/2)**
>
> **Q1: Real-world Deployment**
>
> **A1:** Thank you for your thoughtful and detailed review. The transition from simulation to the real-world deployment is indeed a critical issue. For UAVs, we can leverage the ROS framework to connect with real drones, deploying large models on ground stations to receive sensor data such as IMU and RGB data, and return the expected path to complete the flight. Some work like Air-M[1] has implemented hardware-in-the-loop simulation, supporting the transition to real environments.
>
> However, the exploration of UAV VLN tasks is still limited, and achieving vision-language understanding and long-sequence planning in dynamic environments remains **challenging**. In the latest work, AirealVLN[2] (which gives predefined step-by-step navigational instructions) has an SR of 7.2% on its simple dataset, 2.9% on the full dataset , and 8.69% for CityNav[3] (which gives position of markers near the endpoint in the map). Therefore, we are currently focusing on simulating realistic UAV operations in a simulated environment, aiming to enhance navigation performance before applying it to real-world experiments.
>
> **Q2:** **Autonomy and Reliance on Assistance**
>
> **A2:**  In designing the assistance, we considered the low success rates (<10%) of previous UAV VLN methods and the increased complexity of our task. CityNav uses an aerial perspective, and AirealVLN directly moves the drone's position, reducing the need for obstacle avoidance. CityNav assists navigation by providing the location of landmarks near the endpoint. Therefore, we implemented different levels of assistance. We aim for the highest level of assistance to be like a navigation tool used when driving in an unfamiliar environment, while lower levels provide lighter guidance to help the drone gradually achieve autonomous navigation.
>
> We conducted an ablation experiment without assistance. The experiment showed a sharp decline in performance without assistance in the current long-sequence trajectory search tasks. Notably, our L3 setup only uses the drone's depth camera sensors and does not rely on external information like GT trajectories, allowing the drone to autonomously navigate and complete tasks. However, completing long-trajectory VLN tasks without guidance still presents challenges.
>
> | **Assist Level** | **SR** | **OSR** | NE     | SPL   |
> | ---------------- | ------ | ------- | ------ | ----- |
> | No assist        | 6.25   | 11.88   | 167.10 | 4.95  |
> | L3 assist        | 7.20   | 13.71   | 163.55 | 6.17  |
> | L1 assist        | 24.06  | 43.13   | 130.57 | 20.42 |
>
> **Q3:** **Efficiency and Real-time Performance**
>
> **A3:** Thank you for raising this point about efficiency and real-time performance. Indeed, the complexity of LLMs makes real-time deployment on resource-limited drones challenging. A more feasible approach is to deploy the LLM-based navigation model on **the drone's ground station**, while traditional flight control algorithms run on the drone itself. They exchange multimodal perception data and flight paths via a communication link. In this setup, the real-time performance of drone navigation depends on the inference and transmission latency of the LLM navigation algorithm.
>
> Testing shows that using a bf16 precision on a single 4090 GPU, the average inference time of the navigation model is **0.407 seconds**. With techniques like model quantization, this time can be further reduced. **The model can predict waypoints for the drone to fly 5 meters ahead within 0.5 second.** When the drone flies at a safe speed, such as 2 m/s, the inference time remains shorter than the execution time, ensuring real-time drone navigation.

---

> ### Author Response · Authors · 2024-11-22
> **Response to Reviewer LL2n (Part 2/2)**
>
> **Q4: Ablation Studies**
>
> **A4:** Thank you for pointing this out. Considering the extended time required for full model training and testing, we have supplemented various ablation experiments on a small-scale seen test set. In A2, we provided a performance comparison under different assistance levels.
>
> We further conducted an ablation analysis on the hierarchical trajectory generation model in terms of performance and computational efficiency. The table below shows that using MLLM alone for long-distance prediction results in a decrease in success rate with similar inference times. Fine-grained predictions using MLLM improve performance by receiving assistance information more frequently. However, this comes at the cost of a significant increase in inference time, making it unsuitable for real-time navigation.
>
> | **Model**                          | SR    | OSR   | NE     | SPL   | Time [s]     |
> | :--------------------------------- | ----- | ----- | ------ | ----- | ------------ |
> | Hierarchical trajectory generation | 24.06 | 43.13 | 130.57 | 20.42 | 0.38 + 0.027 |
> | MLLM for long-range predictions    | 20.94 | 40.00 | 137.58 | 17.99 | 0.38         |
> | MLLM for fine-grained predictions  | 25.94 | 50.31 | 126.24 | 22.67 | 1.90         |
>
> We also conducted ablation experiments with fewer sensors. We believe that for search tasks, the front view and top view are more important for detection, so we kept only these two perspectives for the VLN tasks. The experimental results show a slight performance drop across various metrics after reducing the number of sensors, indicating that visual information plays a supportive role in decision-making.
>
> | **Model**               | SR    | OSR   | NE     | SPL   |
> | :---------------------- | ----- | ----- | ------ | ----- |
> | Baseline                | 24.06 | 43.13 | 130.57 | 20.42 |
> | Front and top view | 22.50 | 41.88 | 132.19 | 18.69 |
>
>
>
> **Q5: Comparison with Ground-based VLN**
>
> **A5:** Our method primarily addresses the realism of drone flight. Previous VLN tasks used fixed action sets, positioning agents directly at desired locations, which is reasonable for ground robots. However, drones require attitude adjustments in the air, necessitating waypoint-level output and continuous flight control. Thus, we proposed a waypoint-level UAV navigation model that uses hierarchical output to balance performance and efficiency, aiming to better align with real UAV VLN tasks.
>
> Some classic ground-based VLN methods, like Seq2Seq, can be adapted to our task, but they must be guided to produce waypoint-level outputs. In contrast, models like DUET and NavGPT2 focus on discrete VLN tasks, relying on topological navigation graphs for decision-making, which significantly differs from our task and poses application challenges.
>
> Therefore, we trained a Seq2Seq network and tested it using a small batch of test data. The experimental results indicate that understanding multimodal information remains a challenge for Seq2Seq.
>
> | Method   | SR    | OSR   | NE     | SPL   |
> | -------- | ----- | ----- | ------ | ----- |
> | Seq2Seq  | 6.56  | 13.13 | 171.42 | 5.32  |
> | Baseline | 24.06 | 43.13 | 130.57 | 20.42 |
>
>
>
> **Q6: Extensibility to Other Tasks**
>
> **A6:** Thank you for noting the extensibility of the platform and dataset. Yes, we have made target search one of the tasks supported by the platform. The platform offers a rich set of object assets, interfaces for scene object setup, user-defined scene customization, drone control, and asynchronous parallel data collection, making it ready to expand to more UAV tasks. The continuous data collection framework with equal time intervals and closed-loop simulation support facilitates training UAV trajectory planning models and end-to-end large models.
>
> Our dataset supports target-oriented drone tasks, with drone trajectories that can be reused for various tasks by applying targeted language annotations. For example, annotating UAV action commands to decompose drone trajectories and train low-level instruction-to-trajectory models.
>
> We have also collected depth maps from multiple drone camera perspectives within the dataset, which can be used for mapping tasks. Additionally, using the platform's asynchronous data collection feature, more sensor data, such as LiDAR point clouds, can be collected based on existing trajectories.
>
>
>
> **References:**
>
> [1] Air-M: A Visual Reality Many-Agent Reinforcement Learning Platform for Large-Scale Aerial Unmanned System
>
> [2] AerialVLN: Vision-and-Language Navigation for UAVs
>
> [3] CityNav: Language-Goal Aerial Navigation Dataset with Geographic Information

---

> > ### Comment · Reviewer_DUG4 · 2024-11-22
> >
> > Thanks for your rebuttal; it solves most of my confusion. I will maintain the positive score and recommend the acceptance of this paper.

---

> ### Author Response · Authors · 2024-11-26
> **Response to Reviewer LL2n**
>
> Dear Reviewer LL2n,
>
> Thank you for your valuable suggestions on our manuscript. Based on your comments and questions in the previous round of review, we have provided additional clarifications on the following aspects: Real-world Deployment, Autonomy and Reliance on Assistance, Efficiency and Real-time Performance, Ablation Studies, Comparison with Ground-based VLN, and Extensibility to Other Tasks.
>
> We would greatly appreciate it if you could consider our further explanations and revisions.
>
> Best regards,
>
> Authors of Paper 611

---

> > ### Comment · Reviewer_LL2n · 2024-11-26
> >
> > I appreciate the author's responses. The authors address most of my concerns. I still maintain the positive score and hope the authors revise the paper according to the reviewers' reviews.

---

> > > ### Author Response · Authors · 2024-11-27
> > > **Response to Reviewer LL2n**
> > >
> > > Dear Reviewer LL2n,
> > >
> > > Once again, thank you for the time and effort you have dedicated to reviewing our paper. We are pleased to hear that most of your concerns have been addressed.  Based on your suggestions and the feedback from other reviewers, we have improved the clarity of the paper’s expressions and added more comprehensive ablation experiments. We truly appreciate your valuable suggestions, which have contributed to enhancing the quality of our revised version.
> > >
> > > Best regards,
> > >
> > > Authors of Paper 611

---

### Official Review · Reviewer_DUG4 · 2024-11-04

**Soundness:** 3
**Presentation:** 3
**Contribution:** 3
**Rating:** 6
**Confidence:** 3

**Summary:**

This paper makes three key contributions to advance realistic UAV vision-language navigation. First, they introduce OpenUAV, a simulation platform that properly models realistic drone flight dynamics with 6 degrees of freedom, rather than using simplified discrete movements like previous work. Second, they create what appears to be the first UAV navigation dataset (about 12k trajectories) that incorporates realistic flight physics. Third, they propose a novel "UAV-Need-Help" benchmark where an AI assistant provides varying levels of guidance, along with a hierarchical navigation model using large language models.
The approach significantly outperforms baselines in experiments, though still falls well short of human performance. What makes this work valuable is how it addresses fundamental limitations in current UAV navigation research by developing tools and datasets that better match reality, particularly through the inclusion of realistic flight dynamics. While substantial work remains before real-world deployment would be feasible, this paper lays important groundwork and provides valuable resources for future research in autonomous drone navigation.

**Strengths:**

1. The paper address realistic flight dynamics in UAV navigation instead of using oversimplified discrete actions.
2. Provides a comprehensive open-source simulation platform that combines realistic environments, flight physics, and algorithmic support.
3. Creates the first large-scale UAV navigation dataset (12k trajectories) that captures true 6-DOF flight dynamics.
4. Introduces a novel assistance mechanism with three levels of guidance, making the navigation task more adaptable and learnable.
5. Proposes a practical hierarchical architecture combining language models with trajectory planning for complex navigation.
6. Includes thorough experimental validation across different difficulty levels and scenarios, with clear comparisons to human performance.
7. Shows strong generalization ability on unseen objects, though with expected performance drops on unseen environments.

**Weaknesses:**

1. The dataset size (12k trajectories) is relatively small compared to other vision-language datasets, which might limit model learning.
Lacks detailed comparison with real drone flight data to validate how well their simulation matches actual flight dynamics.
2. No discussion of computational requirements or real-time performance, which is crucial for practical drone navigation.
3. Success rate metrics (around 16-17%) are still quite low even with the highest level of assistance.
4. Missing ablation studies on different environmental conditions and their impact on navigation performance.
5. No discussion of safety mechanisms or fallback strategies when the navigation system fails.

**Questions:**

Please see the weakness.

---

> ### Author Response · Authors · 2024-11-22
> **Response to Reviewer DUG4 (Part 1/2)**
>
> **Q1:** **Size of the dataset and authenticity of the simulated data**
>
> **A1:** Thank you for noting our simulation dataset. We believe that the current dataset is sufficient to support the training of MLLM-base models. We would like to address your concerns as follows:
>
> - In Table 1, we compare our dataset with mainstream vision-language navigation datasets. While the total number of trajectories in our dataset is **comparable** to current VLN benchmarks, our trajectories contain the **highest number of actions** per sequence, represented as detailed drone trajectory points.
> - We segment each full trajectory into **shorter training trunks** during model training at every timestep. This method results in 405k trunks.
> - We train our model on a **pre-trained** foundation model with basic vision-language capabilities, utilizing LoRA fine-tuning. This approach requires less data compared to training from scratch.
>
> Regarding the authenticity of the data, during data collection, we established a control chain connecting the remote controller, QGC, PX4, and the simulation environment, aligning it with the control chain used in real-world drone operations. Furthermore, our data was collected entirely by human pilots rather than generated using algorithms such as shortest-path planning, ensuring its authenticity.
>
> **Q2:**  **Efficiency and real-time performance**
>
> **A2:** Thank you for raising this point about practical drone navigation. We agree that further investigation would be valuable, particularly regarding trade-offs between inference speed and model performance.
>
> After testing, the average inference time for a single forward pass of the model using bf16 precision on a single NVIDIA 4090 GPU is **0.407 seconds**. If model quantization or other acceleration techniques are applied, the inference time can be further reduced. **The model is capable of predicting the trajectory points for a 5-meter flight within 0.5 second**. At a safe drone flight speed, such as 2 m/s, the model's inference time is shorter than the execution time, ensuring real-time navigation.
>
> Additionally, we conducted an ablation study on the hierarchical trajectory generation model by sampling a subset of seen test set, considering the extended time required for full testing. As shown in the table below, we compared the model's performance and computational efficiency.
>
> | **Model**                          | SR    | OSR   | NE     | SPL   | Time [s]     |
> | :--------------------------------- | ----- | ----- | ------ | ----- | ------------ |
> | Hierarchical trajectory generation | 24.06 | 43.13 | 130.57 | 20.42 | 0.38 + 0.027 |
> | MLLM for long-range predictions    | 20.94 | 40.00 | 137.58 | 17.99 | 0.38         |
> | MLLM for fine-grained predictions  | 25.94 | 50.31 | 126.24 | 22.67 | 1.90         |
>
> The results indicate that using MLLM alone for long-range predictions results in a slight decrease in success rate with similar inference times. Fine-grained predictions using MLLM improve performance by receiving assistance information more frequently. However, this comes at the cost of a significant increase in inference time, making it unsuitable for real-time navigation.
>
> **Q3:** **Concern about success rate metrics**
>
> **A3:** Currently, the exploration of UAV VLN tasks is still limited, and achieving vision-language understanding and long-sequence planning in dynamic environments remains **challenging**. In the latest work, AirealVLN[1] (which gives predefined step-by-step navigational instructions) has an SR of 7.2% on its simple dataset, 2.9% on the full dataset , and 8.69% for CityNav[2] (which gives position of markers near the endpoint in the map), that's why we add assistance in this task.
>
> On the other hand, our model **achieves an OSR of 44%**, indicating that the model is capable of reaching areas near the destination with assistance. However, due to limitations in detection accuracy, the model struggles to identify targets such as humans, animals, and small objects like road signs. Compared to the latest UAV VLN task, we have achieved a great improvement in OSR with the assistance.

---

> ### Author Response · Authors · 2024-11-22
> **Response to Reviewer DUG4 (Part 2/2)**
>
> **Q4:** **Ablation studies on different conditions**
>
> **A4:** Thank you for raising this question about performance under different scenarios and target categories. We resampled the test set from the perspectives of scenarios and objectives to evaluate drone navigation performance under different conditions, yielding the following experimental results.
>
> | **Scenario**   | **SR** | **OSR** |
> | -------------- | ------ | ------- |
> | TropicalIsland | 26.22  | 77.44   |
> | ModernCityMap  | 24.79  | 60.33   |
> | Carla_Town01   | 23.78  | 54.88   |
> | NewYorkCity    | 5.52   | 23.45   |
>
> Open natural environments like TropicalIsland experience fewer collisions and have a higher success rate. Smaller maps such as Carla_Town01 and ModernCityMap, also achieve high success rates. However, complex environments remain highly challenging. For example, the narrow streets of NewYorkCity, with higher buildings on both sides and obstacles like streetlights and power lines, result in lower success rates.
>
> | **Target** | **SR** | **OSR** |
> | ---------- | ------ | ------- |
> | Car        | 32.86  | 51.43   |
> | Human      | 14.29  | 42.86   |
>
> Testing two common target objects on the easy (less than 250 meters) test set of the large city map NYCEnvironmentMegapa showed that the salience of the target significantly affects the success rate (32.86% for cars and 14.29% for humans).
>
> **Q5:** **Safety mechanism of navigation system**
>
> **A5:** Thank you for noting the safety mechanisms. The main contribution we claimed in this paper is the UAV simulation platform and the assistant-guided UAV object search benchmark, focusing on advancing UAV visual-language navigation tasks. However, we agree with your suggestion. In our closed-loop simulation, we employ strategies such as hovering or immediate landing in response to severe navigation failures like collisions. Besides, we need to further investigate safety mechanisms to enhance the navigation system during experimental validation involving physical UAVs.
>
> **References：**
>
> [1] AerialVLN: Vision-and-Language Navigation for UAVs
>
> [2] CityNav: Language-Goal Aerial Navigation Dataset with Geographic Information

---

> > ### Author Response · Authors · 2024-11-23
> > **Response to Reviewer DUG4**
> >
> > Dear Reviewer DUG4,
> >
> > Thank you once again for taking the time to review our paper and for participating in the rebuttal discussion. We are glad to hear that you found our response helpful.
> >
> > Best regards,
> >
> > Authors of Paper 611

---

### Author Response · Authors · 2024-11-22
**General Response**

Dear AC and Reviewers,

Thank you very much for your valuable comments and insightful suggestions, which have helped us improve the quality of our original manuscript. We have provided detailed discussions and updated our manuscript based on the reviewers' concerns. Below is a summary of the work we have done during the rebuttal phase.

We have enhanced the readability of the revised manuscript. For example, we modified the platform naming and function descriptions, clarified the introduction of related work and the inaccurate statements in the model structure, and provided more details on the specific implementation of DAgger. We have made these revisions based on the reviewers' suggestions to more clearly present the platform, benchmark, and method we proposed.

We have supplemented more experimental details. Considering the time required for complete training and evaluation, we sampled part of the test data for quantitative experiments, focusing on the following aspects: 1) model structure analysis, 2) autonomous navigation without assistance, 3) performance of other VLN models. We hope that the additional experimental results can address the reviewers' concerns.

We further discussed our motivations and innovations, highlighting the realistic continuous trajectory tasks supported by the platform, multi-level assistance that enables gradual autonomy for UAVs in visual-language navigation, and the hierarchical framework of small and large models that balances performance and efficiency.

We emphasized the challenges of UAV VLN tasks, which are still in the early exploration stage, and expressed our hope that through our platform, benchmark, and method, we can narrow the gap with real UAV flights, providing a safe, controllable, and easy-to-use environment to support better performance and more autonomous UAV VLN model training, as well as the goal of transferring VLN tasks to real-world scenarios.

Finally, we would like to thank the four anonymous reviewers once again for their insightful comments and valuable suggestions, which have greatly helped us improve the technical quality and presentation of the manuscript. We also thank the Area Chair for reading our responses again.

Sincerely,

Authors of Paper 611

---

### Meta-Review · Area_Chair_6Ff3 · 2024-12-21

**Metareview:**

The paper targets UAV-based vision-language navigation, recognizing that drones move in six degrees of freedom and therefore require continuous control, unlike ground-based robots. It presents the OpenUAV simulation platform, which offers realistic flight physics and varied environments. A dataset of around 12,000 drone trajectories is introduced, designed specifically for UAV navigation using language instructions. The authors also propose the UAV-Need-Help benchmark, which uses different levels of assistance to guide UAVs more effectively in challenging scenarios. To handle these tasks, they develop a hierarchical large language model that processes both images and text, then generates high-level navigation goals and detailed flight paths.

Strengths of the Paper:

-- Realistic UAV Flight Dynamics: By implementing full 6-DoF movement and continuous control, the paper tackles the limitations of simplified or discrete motion in previous UAV navigation research.

-- Comprehensive Toolkit: The authors provide a cohesive suite—including the OpenUAV platform, a large-scale dataset, and a structured benchmark—that can accelerate progress in UAV VLN research.

-- Hierarchical Model Design: Their hierarchical approach leverages LLM capabilities for high-level planning and detailed trajectory generation, demonstrating strong performance improvements compared to baseline methods.

Weaknesses of the Paper:

-- Limited Real-World Validation: There is no direct comparison with real-flight data or physical drones to confirm that the simulated environment’s flight dynamics transfer effectively to real-world conditions.

-- Insufficient Ablation and Comparison: While the authors mention a hierarchical LLM-based approach, the paper lacks thorough ablation studies (e.g., the impact of removing hierarchical layers, reliance on assistance levels) and deeper comparisons with related UAV VLN platforms or ground-based VLN methods.

-- Scalability and Real-Time Concerns: The paper does not adequately discuss the computational overhead, latency, or hardware constraints that might limit real-time deployment, an essential factor for practical UAV applications.

Overall, the paper provides a substantial contribution to UAV vision-language navigation research, and the reviewers all agree on accepting this paper. The Area Chair recommends accepting the paper.

**Additional Comments On Reviewer Discussion:**

The weaknesses are described above. The authors have addressed most comments in rebuttal and the reviewers generally agree to accept the paper.

---

### Decision · Program_Chairs · 2025-01-22

Accept (Poster)